

# Basic and extensible post-processing of eddy covariance flux data with REddyProc.

Thomas Wutzler[1], Antje Lucas-Moffat[2,3], Mirco Migliavacca[1], Jürgen Knauer[1], Kerstin Sickel[1], Ladislav Šigut[4], Olaf Menzer[5], and Markus Reichstein[1]

[1]Max Planck Institute for Biogeochemistry, Hans-Knöll-Straße 10, 07745 Jena, Germany
[2]German Meteorological Service, Centre for Agrometeorological Research, Bundesallee 33, 38116 Braunschweig, Germany
[3]*Also at:* Thuenen Institute of Climate-Smart Agriculture, Bundesallee 65, 38116 Braunschweig, Germany
[4]Global Change Research Institute CAS, Bělidla 986/4a, CZ-60300 Brno, Czech Republic
[5]Department of Geography, University of California, Santa Barbara, CA 93106-4060, USA

**Correspondence:** Thomas Wutzler
(twutz@bgc-jena.mpg.de)

**Abstract.**

With the eddy-covariance (EC) technique, net fluxes of carbon dioxide ($CO_2$) and other greenhouse gases as well as water and energy fluxes can be measured at the ecosystem level. These flux measurements are a main source for understanding biosphere-atmosphere interactions and feedbacks by cross-site analysis, model-data integration, and up-scaling. The raw fluxes
measured with the EC technique require an extensive and laborious data processing. While there are standard tools available in open source environment for processing high-frequency (10 or 20 Hz) data into half-hourly quality checked fluxes, there is a need for more usable and extensible tools for the subsequent post-processing steps. We tackled this need by developing the `REddyProc` package in the cross-platform language R that provides standard $CO_2$-focused post-processing routines for reading (half-)hourly data from different formats, estimating the uStar threshold, gap-filling, flux-partitioning, and visualizing the
results. In addition to basic processing, the functions are extensible and allow easier integration in extended analysis than current tools. New features include cross year processing and a better treatment of uncertainties. A comparison of `REddyProc` routines with other state-of the art tools resulted in no significant differences in monthly and annual fluxes across sites. Lower uncertainty estimates of both uStar and resulting gap-filled fluxes with the presented tool was achieved by an improved treatment of seasons during the bootstrap analysis. Higher estimates of uncertainty in day-time partitioning resulted from a better accounting of
the uncertainty in estimates of temperature sensitivity of respiration. The provided routines can be easily installed, configured, used, and integrated with further analysis. Hence the eddy covariance community will benefit from using the provided package, allowing easier integration of standard post-processing with extended analysis.





## 1 Introduction

The availability of ecosystem level observations of net ecosystem exchange (NEE) of carbon dioxide ($CO_2$) and other gases, latent heat (LE) and sensible heat (H) fluxes measured by the eddy covariance (EC) method (Aubinet et al., 2000) allowed a boost in ecosystem understanding at site to global scales (Baldocchi et al., 2017). The EC method provides half-hourly or

hourly records of turbulent fluxes between an entire ecosystem and the atmosphere. These data are derived from high frequency measurements (10 or 20 Hz) of wind speed and direction together with measurements of air scalar characteristics such as $CO_2$ and water vapor concentration, or temperature. Methods to compute fluxes from high frequency measurements have been consolidated in the last decades (Rebmann et al., 2012; Foken et al., 2012; Aubinet et al., 2012). Depending on the site characteristics, the NEE fluxes also need to be storage corrected. Although measured continuously, the (half)hourly EC data

contain gaps due to instrument malfunction or meteorological conditions under which the assumptions of the EC technique are not met. Such conditions include insufficiently developed turbulence or low signal stability (details in e.g. Foken and Wichura, 1996; Foken et al., 2004; Göckede et al., 2004; Foken et al., 2012). Data obtained under these unfavorable conditions are subject to several constraints, technically correct but not representative of the ecosystem. Hence, (half-)hourly records are flagged for different quality levels and need further extensive post-processing as described by Papale et al. (2006).

NEE from periods with low friction velocity (uStar) (Aubinet et al., 2012) need to be detected and filtered out to avoid systematic biases in nighttime NEE (Papale et al., 2006). The screened flux time series with gaps need to be filled (Reichstein et al., 2005a) using the available flux data and auxiliary micrometeorological measurements. An additional information can be obtained from NEE thanks to flux partitioning methods, that provide model estimates of gross primary production (GPP) and ecosystem respiration ($R_{eco}$) (Reichstein et al., 2005a). These gross fluxes are important to understand land-atmosphere interactions.

All these post-processing steps need to be performed routinely for EC data. Hence, it is desirable to have automated and reproducible post-processing tools available that can be easily used, extended, and integrated into researcher's own workflow. For this purpose we compiled all routines for the important $CO_2$-focused post-processing steps in the `REddyProc` package for the cross-platform, free R language. The `REddyProc` package loads time series of quality checked and storage corrected fluxes and the basic set of meteorological variables and provides a software environment to perform uStar threshold detection and

filtering, gap filling and partitioning. Furthermore, a series of other functionalities like data import routines and data visualization are provided.

The objectives of the paper are to 1) provide a reference that describes the methodology of the processing used in the `REddyProc` package, and 2) show that the obtained results do not differ systematically from results obtained with standard post-processing implemented in the FLUXNET community (based on Papale et al., 2006; Reichstein et al., 2005a; Lasslop et al.,

2010; Pastorello et al., 2017).

Table 1 explains used abbreviations. The first part of the paper (section 2) describes the post-processing methods. The second part (Section 3) presents the benchmarks the `REddyProc` implementation with standard post-processing tools. It details





differences in the implementations and possible consequences in obtained results and aggregated fluxes. An appendix gives an overview of the package with general design, an example of the post-processing, and links to help and resources so that readers can get started post-processing their own data.

**Table 1.** Abbreviations used repeatedly in the paper.

| Symbol | Description |
|---|---|
| EC | eddy-covariance |
| $CO_2$ | carbon dioxide |
| NEE | net ecosystem exchange towards the atmosphere in $\mu mol\,CO_2\,m^{-2}s^{-1}$ (aggregated in $gC\,m^{-2}yr^{-1}$) |
| GPP | gross primary productivity (same units as NEE) |
| $R_{eco}$ | ecosystem respiration (same units as NEE) |
| H, LE | sensible and latent heat in $Wm^{-2}$ |
| uStar | friction velocity in $ms^{-1}$ |
| Rg | shortwave incoming global radiation in $Wm^{-2}$ |
| Tair | air temperature in °C |
| VPD | vapor pressure deficit in $hPa$ |
| MDS | marginal distribution sampling (section 2.2) |
| LUT | look-up table (section 2.2.1) |
| MDC | mean diurnal course (section 2.2.2) |
| $E_0$ | temperature sensitivity parameter in eq. 1 |
| $R_{Ref}$ | respiration at reference temperature parameter eq. 1 |
| DP06 | C-implementation of the uStar threshold estimation by Dario Papale (section 3) |
| LRC | light response curve (section 2.3.2) |

## 2 Methods of Post-processing

5 The post-processing relies on half-hourly or hourly measurements of NEE and ancillary meteorological data of uStar, Rg, Tair or soil temperature, and VPD. The fluxes should be quality checked and, if applicable, storage corrected before their use in the package.

The post-processing follows a specific workflow:

1. determination and filtering of periods with low turbulent mixing (uStar-filtering),

10 2. replacing missing data in the half-hourly/hourly records (gap-filling), and



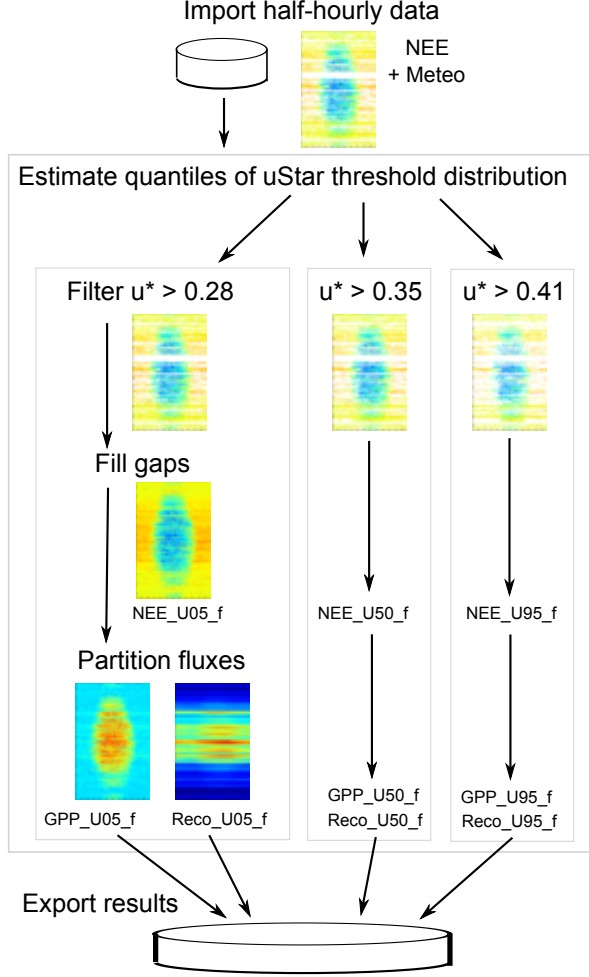

**Figure 1.** The workflow starts with importing the half-hourly (or hourly) data, here the year 1998 of site DE-Tha is plotted as an example. Next, a probability distribution of uStar threshold is estimated for each season. Gap-filling and flux-partitioning is performed for several quantiles of this distribution for an estimate of uncertainty. Finally the results are exported.

3. partitioning NEE into the gross fluxes GPP and $R_{eco}$ (flux-partitioning).

Usage of the `REddyProc` package follows this data post-processing workflow (Fig.1). The following sections explain the steps in more detail.

## 2.1 uStar-filtering

5  Determining periods with low turbulent mixing is a critical step in the EC data post-processing. Standard steady state and integral turbulence characteristics tests in the initial processing exclude the most problematic records of sensible (H), latent (LE)




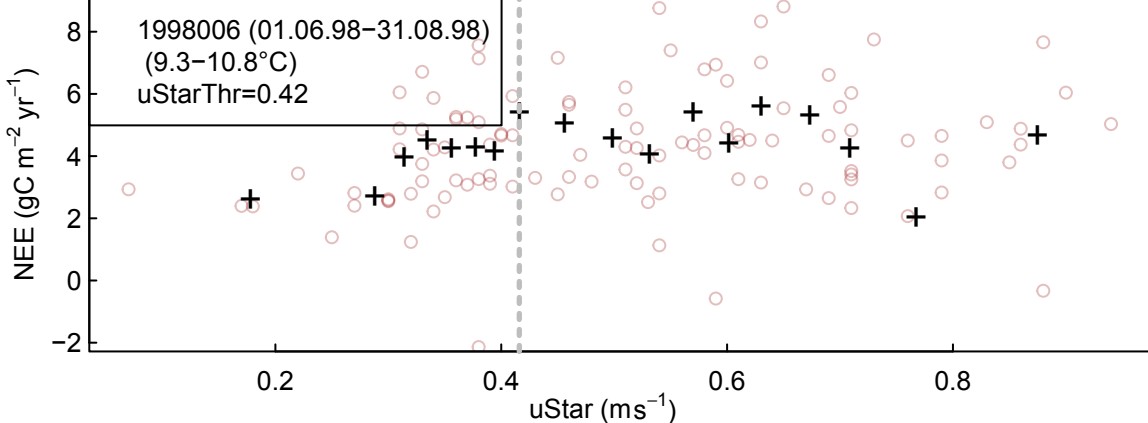

**Figure 2.** Concept of the uStar-filter: NEE at low uStar friction velocities below a threshold, i.e with low turbulence, is biased low compared higher uStar with otherwise similar environmental conditions. The uStar threshold (dashed line) is estimated by a moving point method on uStar bins (crosses) across half-hourly records (circles) here for a season-temperature subset of data from DE-Tha.

heat and $CO_2$ fluxes (Foken and Wichura, 1996). However, it is well known (summarized in Aubinet et al., 2012, chapter 5), that such a quality checking strategy is not sufficient, especially in the case of $CO_2$. Stable stratification that is present often during the night-time dampens turbulence and leads to an underestimation of the night-time NEE, i.e. the ecosystem respiration (Gorsel et al., 2007). Massman and Lee (2002) proposed that unfavorable conditions could be detected by inspecting the relationship of night-time NEE versus uStar. Within similar time period and similar environmental conditions respiration should not be dependent on the uStar. At low uStar values, a negatively biased respiration is measured. A heuristic class of methods, which is widely accepted, assumes that a threshold of uStar can be established above that nighttime fluxes are considered valid. Hence, the uStar threshold is the minimum uStar above which respiration reaches a plateau (Fig. 2). This threshold is specific for each season of a site year. Uncertainties in the uStar threshold estimate represent one of the largest uncertainty components in the post-processing of NEE.

There are at least two methods of estimating the uStar threshold: the moving point method (Reichstein et al., 2005c; Papale et al., 2006), which is currently more routinely used, and the break-point detection method (Barr et al., 2013).

### 2.1.1 Moving point method for uStar

The method of (Papale et al., 2006) detects a plateau in the relationship of night-time NEE versus uStar among all records within temperature subset by a moving point test of records binned into different uStar bins.

The nighttime data (default: Rg < 10 $\mathrm{Wm}^{-2}$) is split into different times of year, here called seasons, to account for differing surface roughness. Then the data of each season is split into default six temperature subsets of equal size (according to quantiles). Within each temperature subset data is split into 20 about equally sized uStar bins. The default moving point method, called `Forward2`, determines the threshold based on these uStar bins. It checks for each bin if the mean NEE is higher than 0.95

times the mean of the following 10 bins. If this holds true also for the next bin, the mean uStar of the bin is reported as threshold. There are often subsets of data, where no clear threshold can be detected. Hence, there are quality criteria on whether the estimate of a given subset is used in subsequent aggregation. One quality criterion specifies that temperature and uStar should not be correlated within the temperature subset, another requires a minimum number of valid records within a subset. Next, the uStar estimates for different temperature classes and periods are aggregated to derive a robust uStar estimate. Within one season,

the median is taken across the estimates of different temperature subsets. Within one year, the maximum is taken across the associated seasons.

Records during night-time with uStar smaller than the estimated threshold are flagged as invalid and are replaced in the subsequent gap-filling processing step.

### 2.1.2 Breakpoint detection method for uStar

Alternatively, a breakpoint-detection can be applied to the unbinned data, that avoids the sensitivity of the Moving point method to the specifics of the binning schemes (Barr et al., 2013). `REddyProc` provides this method by estimating the breakpoint based on unbinned records within the seasons/temperature subsets using the `segmented` R-package (Muggeo, 2003, 2008). However, `REddyProc` differs from (Barr et al., 2013) by keeping the same aggregating scheme of seasonal/temperature estimates to annual thresholds as with the Moving point method.

### 2.1.3 Bootstrapping uncertainty of the uStar threshold

Estimates of the uStar threshold are often sensitive to the specifics of the combination of methods and the data, e.g. the binning, minimum number of records within a season or temperature subset, criteria in aggregation, etc. Therefore, bootstrap (re-sampling with replacement) is applied to generate 200 artificial replicates of the dataset and for each replicate the threshold is estimated (Efron and Tibshirani, 1986; Davison and Hinkley, 1997). The 5%, 50% and 95% of the estimates are reported as a range of

threshold estimates. The subsequent post-processing steps of gap-filling and partitioning are then repeated using those different thresholds to propagate the uncertainty of uStar threshold estimation to derived quantities such as annual NEE, GPP and $R_{\mathrm{eco}}$

### 2.2 Gap-filling methods

Filling of gaps in half-hourly NEE data is necessary to obtain complete time series for the calculation of daily averages or balances such as monthly or seasonal sums.



The dataset of half-hourly fluxes after quality checks and uStar-filtering may contain up to 50 percent gaps, sometimes even higher, depending on the site conditions. For the site-year datasets used in this manuscript as a benchmark, the percentage of gaps ranged from on average of 32% before uStar-filtering to 60% or 48% after gap-filling for upper or lower uStar threshold estimate respectively.

The gap-filling method implemented in `REddyProc` is the so-called marginal distribution sampling (MDS) by (Reichstein et al., 2005b). In a comparison with gap-filling methods for net carbon fluxes by (Moffat et al., 2007), this method performed well for the different artificial gap scenarios ranging from single half-hours to several days. Due to its flexibility in dealing with missing meteorological input data and its fast and highly automated routines available as BGC online tool, the MDS gap-filling method has been widely used. The algorithm exploits the covariation of the fluxes with the meteorological variables and the

temporal autocorrelation of the fluxes based on two basic methods, the (moving) look-up table approach (LUT) and the mean diurnal course (MDC), which are described next.

### 2.2.1 Look-up-table for gap-filling

In the look-up table (LUT) approach, the fluxes are binned by the meteorological conditions within a certain time window. The missing value of the flux is then calculated as the average value of the binned records and its uncertainty estimated from their

standard deviation.

The original LUT consisted of fixed periods over a year (Falge et al., 2001), while in the MDS algorithm, the meteorological conditions are sampled with a moving window around the gap to be filled. Within the chosen time window and respective bin, each meteorological variable should not deviate more than a fixed margin to ensure similar meteorological conditions. The default meteorological variables are Rg, Tair, and VPD with default margins of $50\,\mathrm{Wm^{-2}}$, $2.5\,°C$, and $5.0\,\mathrm{hPa}$, respectively.

### 20 2.2.2 Mean diurnal course for gap-filling

The NEE fluxes have a mean diurnal course (MDC) that follows the course of the sun with only respiration (release of $CO_2$) during nighttime and a combination of respiration and photosynthesis during daytime. This autocorrelation of the fluxes is exploited by taking the average value at the same time of day within a moving time window of adjacent days (Falge et al., 2001). Though the MDC method only showed a medium performance in the gap filling comparison by (Moffat et al., 2007), it has the

advantage that this approach can be used even if no meteorological information is available.

In the MDC algorithm, the same time of day includes also the fluxes of the adjacent hour (±1 hour). The number of adjacent days need to be specified.



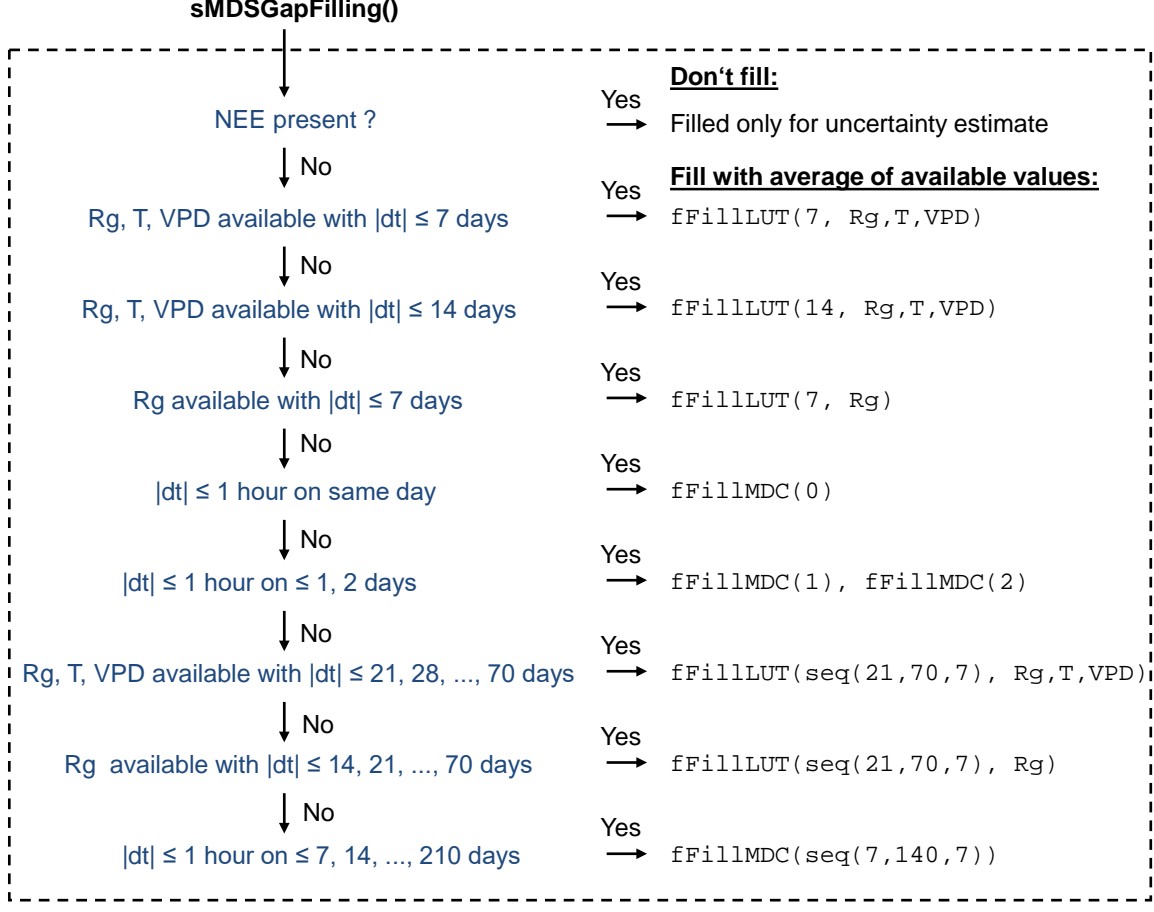

**Figure 3.** Flow diagram of the MDS gap-filling algorithm as implemented in REddyProc. See table 1 for abbreviations.

### 2.2.3 Marginal distribution sampling for gap-filling

The marginal distribution sampling combines the LUT and MDC methods depending on the availability of the meteorological data. Three different conditions are identified for each half-hourly NEE flux:

1. The data of all three meteorological variables (Rg, Tair, and VPD) are available.

2. Tair or VPD are missing, but Rg is available.

3. Also Rg is missing.

Case 1): The missing value is replaced by the average value under similar meteorological conditions in a LUT approach. Similar meteorological conditions based on Rg, Tair and VPD. If no similar meteorological conditions (minimum of two half-hourly





fluxes) are present within the starting time window of 7 days, the windows size is increased to 14 days.

Case 2): The same LUT approach is taken, but similar meteorological conditions can only be defined via Rg within a time window of 7 days.

Case 3): The missing value is replaced with the mean diurnal course (MDC). The number of days start with one day, thus a linear interpolation of available data at adjacent hours (±1 hour) at the same day. The number is then increased to ±1 and ±2 days.

If after these steps the NEE values could not be filled, the procedure is repeated with increased window sizes until the value can
be filled, see flow diagram in Figure 3.

## 2.3 Flux-partitioning methods

The gross fluxes of GPP into the land system and $R_{eco}$ out of the land system are the two opposing parts of NEE: NEE = $R_{eco}$ - GPP. Availability of GPP and $R_{eco}$ is pivotal as they are the two biggest terms of the carbon cycle (e.g. Chapin et al., 2006; Jung et al., 2011). Moreover, understanding their sensitivity to environmental drivers (e.g., radiation, temperature, and soil moisture)
is important to interpret land-atmosphere interactions and to improve earth systems models (Reichstein et al., 2012). Therefore several methods were developed to partition NEE into these two components (Reichstein et al., 2005c; Lasslop et al., 2010; Moffat, 2012; Wehr and Saleska, 2015; Desai et al., 2008; Stoy et al., 2006).

The two most widely used methods are the so-called night-time partitioning and day-time partitioning (Reichstein et al., 2012). The night-time partitioning (Reichstein et al., 2005c) relies on the temperature response function of nighttime NEE fluxes
that are representative of $R_{eco}$. It assumes that this relationship is applicable also to daytime data. The relationship is then used to predict $R_{eco}$ from measured temperature and GPP is computed as a difference between $R_{eco}$ and NEE. This method is currently the most widely used approach. Alternatively, the day-time partitioning (Lasslop et al., 2010) fits a LRC to daytime NEE observations, accounting for the effects of radiation and VPD on GPP as well as the effects of temperature on $R_{eco}$.

### 2.3.1 Nighttime flux-partitioning

The method of (Reichstein et al., 2005c) estimates a temporally varying respiration-temperature relationship from nighttime data. First night-time data is selected by a threshold of Rg < 10 $Wm^{-2}$, which is congruent with the BGC online tool but differs from the 20 $Wm^{-2}$ reported in (Reichstein et al., 2005c).

Next, temperature sensitivity, $E_0$ of the Lloyd and Taylor (1994) relationship (1) is estimated by fitting the model to successive 15 day periods of night-time data, and the resulting $E_0$ series is aggregated to an annual estimate.





$$R_{eco}(T) = R_{Ref} \, exp\left[E_0\left(\frac{1}{T_{Ref}-T_0} - \frac{1}{T-T_0}\right)\right], \tag{1}$$

where $T_0$ is kept constant at -46.02°C Lloyd and Taylor (1994) and where reference temperature $T_{Ref}$ is 15°C, which is congruent with the BGC online tool but differs from the 10°C reported in (Reichstein et al., 2005c). For robustness each fit is repeated on a trimmed data set excluding records with residuals outside the 5%-95% residual distribution. The annual aggregate

is the mean across the 3 valid estimates having the lowest uncertainty in the fit. Single estimates of $E_0$ are considered valid, if there were minimum of 6 records, temperature ranged across at least 5°C, and estimates were inside the range of 30 to 450 K.

Subsequently, the respiration at reference temperature, $R_{Ref}$, is re-estimated from night-time data using the annual $E_0$ temperature sensitivity estimate for 7-day windows shifted consecutively for 4 days. The estimated value is then assigned to the central time-point of the 4 day period and linearly interpolated between periods. Hence, the obtained respiration-temperature

relationship varies across time.

Finally, $R_{eco}$ is estimated for both day- and night-time from the temporarily varying $R_{eco}$ temperature relationship and daytime GPP is computed as $R_{eco}$ - NEE.

### 2.3.2 Daytime flux-partitioning

The method of (Lasslop et al., 2010) models NEE using the common rectangular hyperbolic light-response curve (LRC) (Falge

et al., 2001):

$$NEE = \frac{\alpha \beta R_g}{\alpha R_g + \beta} + \gamma, \tag{2}$$

where $\alpha$ ($\mu mol\,CO_2\,J^{-1}$) is the canopy light utilization efficiency and represents the initial slope of the light-response curve, $\beta$ ($\mu mol\,CO_2\,m^{-2}s^{-1}$) is the maximum $CO_2$ uptake rate of the canopy at infinite Rg, associated with light saturation, and $\gamma$ ($\mu mol\,CO_2\,m^{-2}s^{-1}$) is a term accounting for ecosystem respiration. The hyperbolic light-response curve is modified to account

for the temperature dependency of respiration after Gilmanov et al. (2003) by setting respiration $\gamma$ to the Lloyd and Taylor respiration model (Lloyd and Taylor, 1994) (1). Further, the LRC setting parameter $\beta$ in (2) to an exponential decreasing function (Körner, 1995) at higher VPD values (3).

$$\beta = \begin{cases} \beta_0 \, exp\left[-k(\mathrm{VPD}-\mathrm{VPD}_0)\right] \\ \beta_0 \end{cases}, \tag{3}$$



where the $VPD_0$ threshold is 10 hPa in accordance with earlier findings at the leaf level (Körner, 1995) ignoring potential vegetation specific differences.

Parameter $T_0$ in (1) was fixed as in the nighttime approach (section 2.3.1). Parameter $T_{Ref}$ was fixed in each window to the median temperature within the window. The other parameters $(E_0, R_{Ref}, \alpha, \beta_0, k)$ of the model are estimated by the following

steps. 1) A time varying temperature sensitivity $E_0$ is estimated from night-time data for a window shifted by two days. 2) The $E_0$ estimates are smoothed across successive windows by fitting a Gaussian Process (Rasmussen and Williams, 2006; Menzer et al., 2013) using the `mlegp` R-package that also estimates uncertainty of the smoothed $E_0$. Next, a prior respiration, $R_{Ref}$, for reference temperature $T_{Ref} = 15$ °C is re-estimated from night-time data for each window with smoothed $E_0$. 3) Parameters of the rectangular hyperbolic light-response curve $(R_{Ref}, \alpha, \beta_0, k)$ are fitted using only day-time data and the previously

determined temperature sensitivity $(E_0)$ for each window. 4) Finally, for each NEE record, GPP and $R_{eco}$ are estimated with the parameter set of the previous valid window and the parameters of the next valid window, and the two results are interpolated linearly by the time difference to the window centers. The following paragraphs add necessary technical details to these steps.

In step 1, parameter $E_0$ is estimated for 12 day windows. Only records with temperature above -1 °C are valid for estimation. Reference temperature $T_{Ref}$ in (1) is set to the median temperature of the window in order to decrease correlation between

estimates of $R_{Ref}$ and $E_0$. A missing estimate is reported for non-valid windows with too few valid records (minNRecInDay-Window = 10), non-convergence of the fitting procedure, or an $E_0$ estimate outside the bounds [50,400]. Missing estimates are filled during the smoothing in step 2.

In step 2, the Gaussian Process takes into account the uncertainty of the $E_0$ fit from night-time in each window. However, if the correlation of $E_0$ across subsequent windows is high, the uncertainty is reduced similar as with repeated measurements. The

respiration $R_{Ref}$ for windows where no fit could be obtained is set to the value from the previous valid window. Again, only records with temperature above -1 °C are used.

In step 3, fitting of other parameters is done for each window centered at the same record as the windows of step 1. By default the fit uses the same weak prior on parameters as the BGC online tool (Lasslop et al., 2010). The prior locations are $k = 0.05$, $\alpha = 0.1$, $R_{Ref}$ = night-time estimate, and $\beta$ = range of NEE values, i.e. the difference between 97% and 3% quantile. The prior

uncertainty is $sd_k = 50$, $sd_\beta = 600$, $sd_\alpha = 10$, and $sd_{R_{Ref}} = 80$.

In each daytime fit, the NEE records are weighted according to their variance (Lasslop et al., 2010, eq. 5). In order to avoid unreasonable leverage of records with a very low estimate of NEE uncertainty, the records of the above 0.7 quantile of weights are associated the weight of the 0.7 quantile. This assigns low influence to records with high uncertainty, but avoids the problem of the high leverage with very low estimates of NEE-uncertainty.

There are certain quality criteria and fall-backs during the daytime fitting in order to obtain reasonable fits. If there are too few valid records (minNRecInDayWindow < 10), or the fitting did not converge, or VPD parameter $k < 0$, then the fit is repeated without the VPD effect, because often there are records where VPD is missing but other variables are available. If fitting did not converge or parameter estimate of $\alpha$ is larger than 0.22, the fit is repeated with $\alpha$ fixed to the last valid value of $\alpha$ from fits in





previous windows. If there are still too few records or the fitting was not valid, a missing result is reported for the window. In addition a missing result is reported if estimated $\alpha < 0$ or $R_{Ref} < 0$ or $\beta_0 < 0$ or $\beta_0 > 250$, or if $\beta_0 > 100$ and at the same time estimated standard deviation $sd_{\beta_0} >= \beta_0$ (Table A1 in Lasslop et al., 2010). The Variance-Covariance matrix of the parameter uncertainty is estimated by bootstrapping the day-time fit. In each sample, the prescribed temperature sensitivity $E_0$ is drawn

from a normal distribution with standard deviation estimated in step 2. In this way also the uncertainty of the night-time fit propagates to the uncertainty of the day-time parameters and subsequently to the inferred gross fluxes.

In step 4, variance of the flux estimates are computed based on the Variance-Contrivance matrices obtained in step 3 with each parameter estimate (Lasslop et al., 2010, eq. 6). The two standard deviations based either on the previous and subsequent valid estimates of the Variance-Contrivance matrices are linearly interpolated with respect to the time difference to the estimates.

Note, that contrary to the night-time based flux-partitioning, both GPP and $R_{eco}$ are model predictions and do not add up exactly to observed NEE.

## 3 Benchmarking `REddyProc` post-processing steps

The post-processing steps implementations of `REddyProc` were benchmarked with current widely used post-processing tools. Specifically, `REddyProc` (version 0.8.1) uStar-filtering results were compared with results by a C-implementation

from Dario Papale (Papale et al., 2006), here referred to as DP06. Results of `REddyProc` (version 1.1.2) gap-filling and flux-partitioning were compared with results obtained by the year-2015-version of the web based tool provided by the institute for biogeochemistry, Jena, best described in Reichstein et al. (2005a). The tool is hereafter referred as the old BGC online tool. Annually and monthly aggregated values, here, refer to the mean across all valid values in a month or a year. In the presence of large gaps, these values can differ from real annual or monthly budgets. The manuscript proceeds describing the dataset used for

benchmarking and thereafter sections for the benchmarking of each processing steps. Within each of those sections, subsections describe differences in the code, report the results of benchmarking, and discuss the main consequences.

### 3.1 Dataset used for benchmarking

Data of twenty-five sites with open data policy of the LaThuile FLUXNET dataset[1] were used for benchmarking. The sites are located in different climate zones and belong to a variety of plant functional types (Table 2) to guarantee testing different

conditions (i.e. presence of snow, management such as cuts and crop rotation, sites disturbed) and ecosystem types (e.g. deciduous vs evergreen, grasslands and forests). Site data contained the following variables: NEE already filtered for quality flags (Foken and Wichura, 1996), despiked and uStar filtered (Papale et al., 2006), random error of NEE computed as described by Reichstein et al. (2005a), Tair and soil temperature (Tsoil), Rg, and VPD. Moreover, NEE time series before the uStar-filtering and the uStar data were downloaded from AMERIFLUX and the European Flux Database to test the uStar threshold estimation.

---

[1] www.fluxdata.org





Finally, time series of gap-filled NEE (NEE$_f$), GPP partitioned with the night-time based method (GPP$_{NT}$) (Reichstein et al., 2005a) were downloaded from the LaThuile dataset, while GPP partitioned with the daytime method (GPP$_{DT}$) were computed with BGC online tool.

**Table 2.** Description of the site and site year used for benchmarking `REddyProc`.

[a] Abbreviations for land cover type from International Geosphere-Biosphere Programme (IGBP) classification: CRO: cropland, DBF: deciduous broadleaf forest, EBF: evergreen broadleaf forest, ENF: evergreen needleleaf forest, GRA: grassland, OSH: open shrubland, WET: permanent wetland, WSA: woody savanna.

[b] Abbreviations for climate from Köppen-Geiger classification: Af: equatorial, rainforest; BSh: hot arid steppe; Cfa: humid, warm temperate, hot summer; Cfb: humid, warm temperate, warm summer; Csa: summer dry, warm temperate, hot summer; Dfb: cold, humid, warm summer; Dfc: cold, humid, cold summer.

| Site | Year | Lat | Lon | Land cover[a] | Climate[b] |
|---|---|---|---|---|---|
| CA-NS7 | 2004 | 56.64 | -99.95 | OSH | Dfc |
| CA-TP3 | 2005 | 42.71 | -80.35 | ENF | Dfb |
| CH-Oe2 | 2004 | 47.29 | 7.73 | CRO | Cfb |
| DE-Hai | 2002 | 51.08 | 10.45 | DBF | Cfb |
| DE-Tha | 1998 | 50.96 | 13.57 | ENF | Cfb |
| DK-Sor | 2006 | 55.49 | 11.65 | DBF | Cfb |
| ES-ES1 | 2000 | 39.35 | -0.32 | ENF | Csa |
| ES-VDA | 2005 | 42.15 | 1.45 | GRA | Cfb |
| FI-Hyy | 1998 | 61.85 | 24.29 | ENF | Dfc |
| FI-Kaa | 2001 | 69.14 | 27.30 | WET | Dfc |
| FR-Gri | 2006 | 48.84 | 1.95 | CRO | Cfb |
| FR-Hes | 1998 | 48.67 | 7.06 | DBF | Cfb |
| FR-Lq1 | 2006 | 45.64 | 2.74 | GRA | Cfb |
| FR-Lq2 | 2006 | 45.64 | 2.74 | GRA | Cfb |
| FR-Pue | 2003 | 43.74 | 3.60 | EBF | Csa |
| IE-Dri | 2004 | 51.99 | -8.75 | GRA | Cfb |
| IL-Yat | 2005 | 31.34 | 35.05 | ENF | BSh |
| IT-Amp | 2004 | 41.90 | 13.61 | GRA | Cfa |
| IT-MBo | 2005 | 46.02 | 11.05 | GRA | Cfb |
| IT-SRo | 2001 | 43.73 | 10.28 | ENF | Csa |
| PT-Esp | 2004 | 38.64 | -8.60 | EBF | Csa |
| RU-Cok | 2004 | 70.62 | 147.88 | OSH | Dfc |
| SE-Nor | 1997 | 60.09 | 17.48 | ENF | Dfb |
| US-Ton | 2004 | 38.43 | -120.97 | WSA | Csa |
| VU-Coc | 2002 | -15.44 | 167.19 | EBF | Af |





## 3.2 UStar-filtering: Benchmark with DP06

Estimation of uStar threshold by `REddyProc` using the default moving point method (section 2.1.1) was benchmarked to estimation based on Papale's DP06 C-implementation (Papale et al., 2006). The benchmark applied a boostrap sample of size 60 and recorded lower, median, and upper quantiles of 10%, 50% and 90% instead of the default 5% and 95% based on a larger sample size to save computing time.

The different estimates of the uStar threshold have potential consequences for the inferred fluxes. To explored those consequences, we used the different thresholds to mark gaps, gap-fill the data, and compute the annual NEE based on the gap-filled time series. NEE uncertainty was estimated by the difference between NEE based on lower quantile uStar and NEE based on upper quantile uStar estimate.

### 3.2.1 Differences in code

The biggest difference of `REddyProc` compared to DP06 is that `REddyProc` by default employs seasons that can span across years. With the *'Within one year'* classification option, which is employed also by DP06, records of December are associated to the same season as January and February of the *same* year. With the default *'continuous'* classification, seasons start the same as in DP06 by default in March, June, September, and December. However, December is treated in the same season as January and February of the *next* year to avoid discontinuities at year boundaries. The annual uStar threshold is then applied according to those continuous seasons spanning year boundaries. For example, the processing of 2014 data would by default use data from Winter 2014 (starting in December 2013) to Autumn 2014 (ending in November 2014). `REddyProc` also allows more flexibility with the *'User specified'* classification into seasons as explained below.

There are further slight differences between `REddyProc` and DP06. Both methods bin in a way such that the number of records in each bin is similar. If there are numerically equal uStar values, they are sorted into the same bin, resulting in bins with unequal record numbers. In DP06, less and sometimes no records are sorted into the subsequent bins hampering the moving point detection. Contrary, the binning with `REddyProc` ensures that there are a minimum number of records in all bins. This often results in fewer bins than without numerically equal uStar values. Moreover, differing from the Papale C-implementation, `REddyProc` employs several more quality criteria. First, when comparing the threshold bin to NEE in the following bins, it makes sure that there are least 3 bins to infer a plateau in NEE. Next, when aggregating the thresholds of different temperature classes to season, it ensures that a threshold was found in at least 20% of the temperature classes. For those seasons where no threshold could be determined, the annual estimate is used. When there are too few records within one year, a single season comprising all records is used for threshold estimation.

In difference to DP06, `REddyProc` re-samples data only within seasons instead across the entire year during the bootstrap, in order to protect periods of similar uStar-NEE relationship and to avoid seasonal biases in re-sampling.




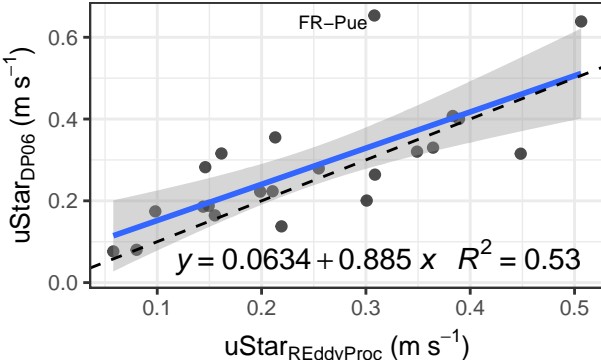

**Figure 4.** Large scatter but retained relationship between uStar derived with DP06 versus uStar derived with `REddyProc` across site-years as shown by a regression (solid line with shaded uncertainty bound) close to the 1:1 line (dashed).

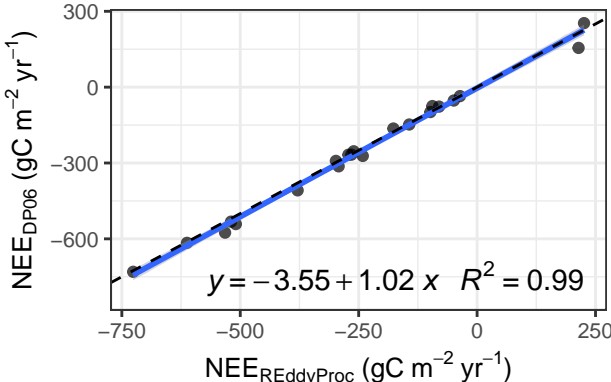

**Figure 5.** Strong correspondence in NEE based on uStar estimated by `REddyProc` and NEE based on uStar estimated by DP06 across site-years.

### 3.2.2 Benchmark results

The general relationship in the estimation of the uStar threshold was retained (slope of 0.94) between the two methods (Fig. 4), although there was a big scatter. The exceptionally high threshold value of $> 0.6\,\mathrm{m\,s^{-1}}$ for site FR-Pue was very probably an overestimate by DP06. However, one has to remember that each estimate has a high uncertainty, and the differences between the two methods were in the range of this uncertainty (Appendix Fig. C1). The estimate of the uncertainty of the uStar thresholds with `REddyProc` was, however, only half of the uncertainty range estimated by DP06 (Appendix Fig. C3). This increased precision was mainly due to the modified bootstrapping scheme, that respects the uStar seasons.

When propagating the differences in uStar to differences in annual NEE, there was no bias and very low scatter across sites between all the methods (Fig. 5), despite the differences in uStar threshold. The absolute differences in annual NEE between



the methods were small (mostly $< 20\,\mathrm{gC\,m^{-2}\,yr^{-1}}$), and mostly lower than half of the uncertainty range estimated from the bootstrap (Appendix Fig. C2). `REddyProc` estimates uStar thresholds with roughly double the precision compared to DP06, due to protecting seasons during bootstrap (Appendix Fig. C4).

### 3.2.3 Discussion of uStar threshold estimation

The agreement between NEE based on uStar estimates of `REddyProc` moving point implementation and current FLUXNET standard post-processing (DP06) (Fig. 5) indicates that the sensitivity of NEE to the uStar threshold estimate in the inferred ranges is low, which also explains the large uncertainty of the uStar threshold estimate. The agreement implies that both methods can be interchanged in studies that are based on aggregated values, such as annual carbon budgets or for upscaling, without the need to reprocess data.

However, the increase in estimated precision, i.e. lower standard deviation, of the uStar threshold estimate also yields an increase in estimated precision of the annual NEE (Fig. C4). This will lead to improved accuracy and usability of EC measurements and any downstream, post-processed data products in model-data integration studies.

### 3.3 Gap-filling: Benchmark with the BGC online tool

The gap-filling implementation of `REddyProc` was benchmarked with the BGC online tool that used pvWave-code from
Reichstein et al. (2005c).

### 3.3.1 Differences in code

Compared to the BGC online tool, the new implementation of the MDS algorithm in `REddyProc` was not limited to single years but filled the gaps with a window moving continuously over all years in the input data. This had the advantage of smoother filling over the end of year time stamp and will especially be of interest for sites which are not dormant during this time. This
new feature led to differences in and probably more realistic gap-filled NEE values in the beginning and end of the year.

There were also slight differences in the window size between the old and new code. For MDC, the old window size had a few more intermediate day steps than the new implementation which affected longer gaps with missing meteorology. The default meteorological variables and margins for LUT (see Chapter 4.2.2 above) were the same in both implementations.

While `REddyProc` restricted gap-filling to interpolation of gaps, the BGC online tool also extrapolated missing records in
periods without measurements.





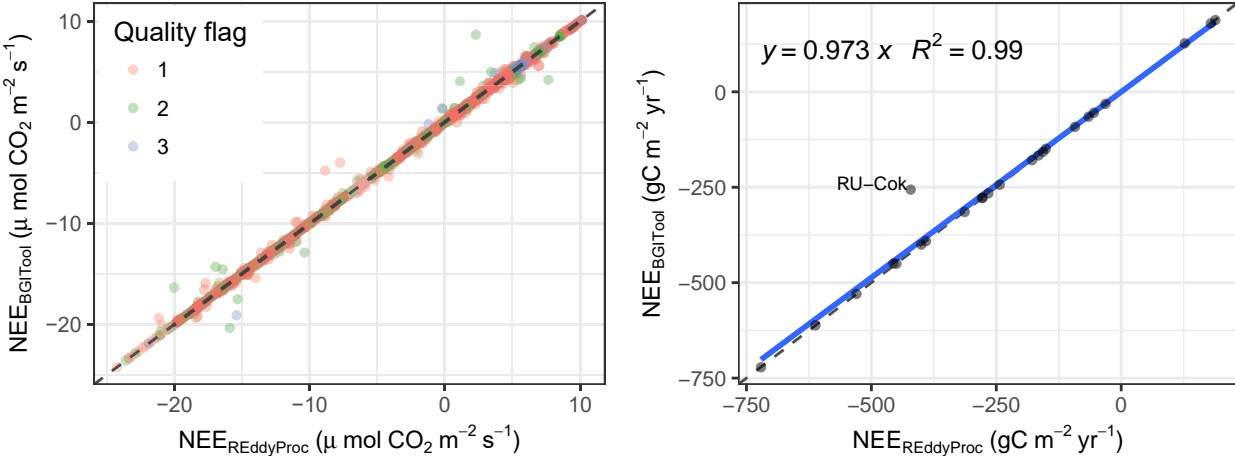

**Figure 6.** Predictions of NEE by `REddyProc` after gap-filling agree with the BGC online tool both at half-hourly values (top), shown for the DE-Tha 1998 use case, and annual means across sites (bottom). Larger quality flags are associated with larger window sizes.

### 3.3.2 Benchmark results for gap-filling

In the benchmark, `REddyProc` gap-filling was run using the same measured NEE as input that passed the QA/QC routines and uStar-filtering. The annually aggregated values comprised both, filled gaps and originally valid records.

`REddyProc` gap-filling results agreed with the results from the BGC online tool. Few discrepancies at half-hourly time
scale were mostly at longer gaps due to usage of fewer window sizes, as shown for the DE-Tha case (Fig. 6 top). At annually aggregated time scale, the agreement between methods was strong ($R^2 = 0.99$) (Fig. 6 bottom). The outlier of site RU-Cok is due to the availability of only a few months of data for the whole year. While `REddyProc` filled gaps in the time period with data available, the old BGC online tool extrapolated also into the time before and after. The seasonal cycle was well reproduced at each site (Appendix C2).

### 3.3.3 Discussion of gap-filling

The good agreement between NEE based on gap-filling by `REddyProc` and the BGC online tool (Fig. 6) imply that these tools can be used interchangeably without the need to reprocess data.

### 3.4 Nighttime flux-partitioning: Benchmark with BGC online tool

The night-time based flux-partitioning was benchmarked to the BGC online tool, that used pvWave code developed by Reichstein
et al. (2005c).





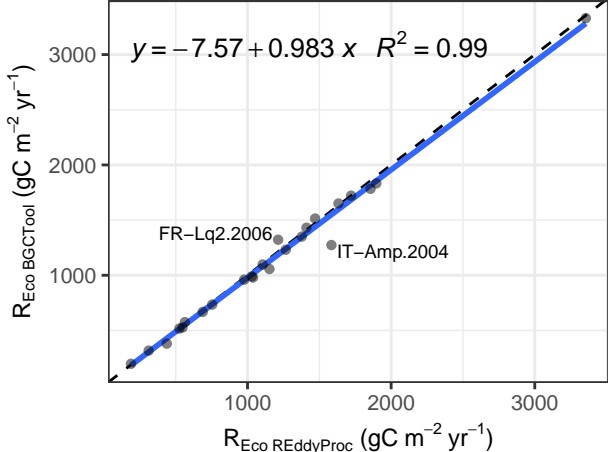

**Figure 7.** Predictions of annually aggregated ecosystem respiration, $R_{eco}$ from `REddyProc` night-time partitioning agree with the predictions by the BGC online tool.

### 3.4.1 Differences in code

The main features of the `REddyProc` implementation of the night-time based partitioning algorithm were very similar to the BGC online tool, with using a reference temperature of 15°C and trimming the estimates of temperature sensitivity $E_0$ before aggregating them (section 2.3.1). `REddyProc` differed from the BGC online tool in computing the potential radiation that is

5    used in subsetting the night-time data to derive $E_0$ and $R_{Ref}$ (Reichstein et al., 2005). While `REddyProc` used the exact solar time for the calculation of the potential radiation, where the sun culminates exactly at noon, the BGC online tool used the local winter time which differs from the solar time depending on the location within the time zone.

### 3.4.2 Benchmark results

Annual aggregated values of $R_{eco}$ predicted by `REddyProc` were in very good agreement ($R^2 = 0.99$; slope $\approx 1$) with the BGC

10    online tool as shown in Fig. 7 and in the Appendix C3.

In order to evaluate the effects of the differences introduced in the code described above, we also computed $R_{eco}$ by prescribing in `REddyProc` either $E_0$, selection of night-time data, or both from the BGC online tool. Results are reported in the Appendix C3 and showed that the most important factor affecting the $R_{eco}$ computed with `REddyProc` was the different selection of night-time data, though the differences were almost negligible at annual time scale (Appendix C3).



### 3.4.3 Discussion of nighttime flux-partitioning

The two implementations agree very well for most sites at annual time-scale. Because of no systematic deviations across sites, the spatial upscaling of fluxes should not be affected by `REddyProc` implementation. However, for some sites, such as IT-Amp, the quite large relative errors indicates problems related to selection of night-time data and problems due to a large gaps in a

dataset.

### 3.5 Daytime flux-partitioning: Benchmark with BGC online tool

The daytime flux-partitioning was benchmarked with results of the BGC online tool. The online tool used pvWave code developed by Lasslop et al. (2010) that was used with slight variations also in the processing of the 2015 Fluxnet release (Pastorello et al., 2017).

### 3.5.1 Differences in code

The BGC online tool differed from `REddyProc` (section 2.3.2) mainly in aspects of separation of nighttime data, estimation of temperature sensitivity from night-time data, uncertainty estimation, treatment of missing values, and optimization library code.

While for separating nighttime data `REddyProc` used exact solar time where sun culminates exactly at noon, the BGC online tool used the local winter time zone time.

For the estimation of temperature sensitivity $E_0$ from night-time data the BGC online tool used a reference temperature of 15 °C, instead of the median temperature inside the window. Hence, it estimated stronger correlations between parameters for windows with a different temperature range. Moreover, it omitted smoothing of the estimated $E_0$ across time, often leading to large fluctuations of the $E_0$ estimates across few days (Fig. C6), larger estimates of its uncertainty, and differences in subsequent estimation of LRC parameters.

For uncertainty estimation, the BGC online tool relied on the curvature of the LRC fit's optimum instead of a bootstrap procedure. Hence, it could not take into account the uncertainty of $E_0$ estimated from night-time data before the daytime LRC fit. Moreover, during interpolation of fluxes based on previous and next valid estimates, the distance weights differed. While `REddyProc` assigned the estimates to the time of the mean of valid record in a window, the BGC online tool assigned it to the start of the third day, also if there were only valid data for the first day in the window.

For weighting the records in the LRC fit, the BGC online tool used the raw estimated NEE uncertainty of each record. It did not check for high leverage of spurious low NEE uncertainty estimates. Its estimates, therefore, were in some windows very strongly influenced by a few records, and failed if a NEE uncertainty estimate of zero was provided. Moreover, when there were missing values or values below zero in provided NEE uncertainty, it set all uncertainty to 1, while `REddyProc` filled the gaps by setting the missing uncertainty to the maximum of 20% of respective NEE but at least 0.7 $\mu mol CO_2 m^{-2} s^{-1}$.





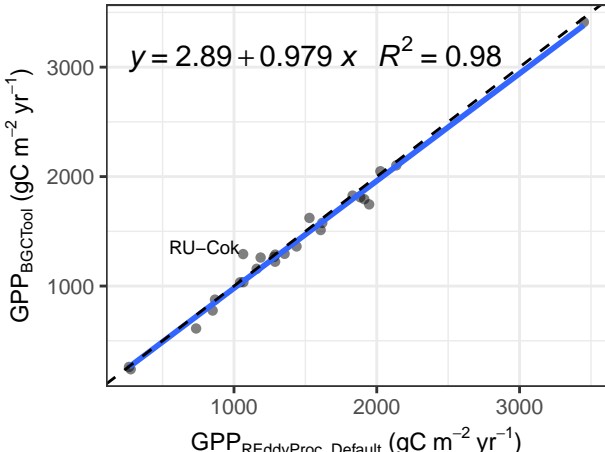

**Figure 8.** Prediction of annually aggregated GPP from `REddyProc` daytime partitioning agree with the BGC online tool across sites.

Treatment of missing values was not considered by the BGC online tool and assumed to be handled prior to the processing. Hence, it did not handle missing VPD values and did not re-try the LRC fit without the VPD-effect in order to use also records with missing VPD. Moreover, as described above, when there were missing values of NEE uncertainty, weighting records in the LRC fit was omitted.

5    For compatibility with the BGC online tool, the above code-differences can be disabled in `REddyProc`. But differences in optimization library code and specifically the conditions of non-convergence on scattered data could not be eliminated, which led to differences in results as shown in the following section.

### 3.5.2    Benchmark results for day-time partitioning

Annually GPP predictions of both implementations showed no significant bias across the test sites (Fig. 8), although, there 10   was some scatter for the individual predictions. A similar scatter was observed when comparing the predictions of the default `REddyProc` options to the predictions with compatibility options. Most of the scatter was caused by skipping the test on high influence of NEE records with small NEE uncertainty (Appendix Fig. C7).

The largest differences in aggregated fluxes between implementations were due to the extrapolation of fitted parameters to periods where no parameter fits were obtained. In many of those cases, there were fits at the boundaries of the difficult periods, 15   whose validity was questionable. Whether those fits passed the quality check or not had a large influence on the extrapolation and hence on the aggregated values. For example, at RU-Cok parameter estimates for valid periods agreed between implementations. However, no valid parameters could be obtained for winter months. While `REddyProc` reported missing values, the BGC online tool reported GPP values based on summer parameterizations also for periods further away from summer.





Uncertainty estimates of gross fluxes were larger with `REddyProc` due to accounting for uncertainty in temperature sensitivity estimate from night-time data (Fig. C5).

### 3.5.3 Discussion of daytime flux-partitioning

Agreement between aggregated fluxes predicted by the daytime method and absence of bias for the test sites (Fig. 8) suggest
that the methods can be used interchangeably for upscaling, although differences in results of influential sites can potentially propagate to differences in upscaled estimates. `REddyProc` provides a quality flag for the results of the day-time partitioning, that allows excluding less reliable data in upscaling studies. For the results associated with good quality flags, we set stronger trust in the `REddyProc`-based estimates.

The daytime flux-partitioning is quite sensitive to the details of LRC fit. Small changes in treatment of extreme or missing NEE
uncertainty estimates or changes in pre-processing and treatment of missing values cause different estimates of LRC parameters and propagate to predicted fluxes of GPP and $R_{eco}$. Although, we put much effort in trying to reproduce the BGC online tool results, we were not able to eliminate all differences, especially in the subtle details in the parameter optimization library codes. The differences in predicted half-hourly fluxes, however, average out across sites and across time (Fig. C8 bottom) making this issue less severe at larger scales.

The estimated uncertainties are even more sensitive. Both implementations occasionally produce unreasonably high outliers that affect the aggregated values. `REddyProc`, in general, estimates higher uncertainties of predicted fluxes, because it accounts for uncertainty in temperature sensitivity. Note, that the uncertainty introduced to annually aggregated fluxes due to flux-partitioning is smaller than uncertainty due to uncertain uStar threshold estimate. Hence, differences or difficulties in uncertainty estimation caused by flux-partitioning do affect conclusions of the overall uncertainty estimates to a lesser extent.

## 4 Conclusions

The `REddyProc` software provides a set of tools for the CO2-focussed post-processing of eddy covariance flux data including uStar-filtering, gap-filling and flux-partitioning. The freely available R-package allows flexible integration into extended workflows and automated routines for the propagation of the uncertainty from the uStar-filtering to the gap-filled NEE and partitioned GPP and $R_{eco}$.

The compatibility of the implemented methods to the available standard tools provides continuity of the data analysis, when adopting `REddyProc` for processing EC-data. `REddyProc` can closely reproduce results of the widely used BGC online tool.

A number of enhancements provide more flexibility to the user in the processing of their data. For instance, the new processing allows to treat multi-year data without breaks at annual boundaries that can significantly affect sites in the southern hemisphere or sites characterized by vegetation activity in winter. Another new feature of `REddyProc` is the flexibility to define different

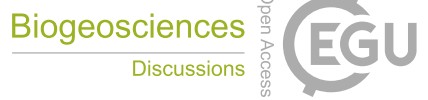

seasons for the application of the uStar-filtering and gap-filling routines, which is important for sites with discontinuous surface cover associated with snow melt, dry seasons, or harvest.

Sensitivity of the results to subtle details of the implementation, however, call for caution when interpreting results. This is especially true for uStar threshold estimation and the daytime-based flux-partitioning, and especially for data with long gaps.

Continued integration of new methodological developments into the package will support research using EC data. We strive to provide new developments in a basic and extensible manner, while paying attention to compatibility with results of reference implementations.

In summary, research using (half-)hourly eddy covariance data can benefit from building blocks for standardized and extensible post-processing provided by `REddyProc`.

**Appendix A: The `REddyProc` package**

The processing tool is freely available in two options: a) online as a web-service[2] with a smaller range of user option, and b) as a package of the open-source R environment with a larger set of user options and with each of the steps and methods available independently.

The `REddyProc` package can be installed by typing at the R-terminal:

```
install.packages("REddyProc")
```
```
library(REddyProc)
```
```
?REddyProc
```

**A1   General design**

There are some general principles and choices in the design of `REddyProc` that explain some trade-offs.

**Preventing accidental errors** is one goal of the package. Therefore, there is extensive checking on formatting and data availability when importing the data and also throughout post-processing steps. Especially, the high-level routines will issue
warnings or stop post-processing if they detect inconsistencies or lack of sufficient data, or changes in specification of critical standard parameters. These checks reinforce a sound standard post-processing also for non-expert users. On the other hand, these checks will render the standard routines not usable for datasets where not enough data are available. It is still possible to use `REddyProc` with sparse data for some purposes by using lower-level routines. But this requires experience in both, post-processing and R programming.

**Smooth inter-annual processing** is the next goal. The data is not partitioned into annual chunks for post-processing to avoid artificial discontinuities between years. The user can still specify different periods or seasons, e.g. when meteorological

---

[2] www.bgc-jena.mpg.de/bgi/index.php/Services/REddyProcWeb





conditions change after harvest, but the boundaries do not need to align with years. REddyProc therefore works with the entire dataset in memory. Potentially this can lead to longer post-processing times when working with many years of EC data but still can be handled on a usual notebook.

**Memory efficient processing** required some extended R programming. Specifically we used R5 classes[3] to avoid frequently
copying the entire dataset in memory. Users should be aware that calling functions on the sEddyProc class not only provides a return value but also changes the data of the R5 class (Chapter 'OO field guide' in Wickham, 2014). Users who want to integrate the post-processing in their own codes are encouraged to learn about R5 classes, however, it is not a prerequisite.

**Continuity with other tools** ensures that switching to REddyProc does not introduce discontinuities with results obtained from other tools. Specifically, we reproduced the uStar threshold estimation of the C implementation by Dario Papale, and the
gap-filling, night-time flux partitioning, and day-time flux-partitioning from the BGC online tool (Reichstein et al., 2005). In some details, the standard parameterization differs from the old tool, e.g. using seasons that span across calendar years in uStar threshold estimation, but it is usually possible to set compatible parameters with REddyProc.

**Appendix B:  Example application**

This section reports an example R session using REddyProc. Code is shown in a shaded area and corresponding output with
monospace font.

**B1   Importing the half-hourly data**

The workflow starts with importing the half-hourly data. The example reads a text file with data of the year 1998 from the Tharandt site and converts the separate decimal columns year, day, and hour to a POSIX timestamp column. Next, it initializes the sEddyProc class.

```r
#+++ load libraries used in this vignette
library(REddyProc)
library(dplyr)
#+++ Load data with 1 header and 1 unit row from (tab-delimited) text file
fileName <- getExamplePath('Example_DETha98.txt', isTryDownload = TRUE)
EddyData.F <- if (length(fileName)) fLoadTXTIntoDataframe(fileName) else
  # or use example dataset in RData format provided with REddyProc
  Example_DETha98
#+++ Add time stamp in POSIX time format
EddyDataWithPosix.F <- fConvertTimeToPosix(EddyData.F, 'YDH',Year.s = 'Year'
```

---

[3] www.rdocumentation.org/packages/methods/versions/3.4.3/topics/ReferenceClasses





```
    ,Day.s = 'DoY',Hour.s = 'Hour')
#+++ Initalize R5 reference class sEddyProc for post-processing of eddy data
#+++ with the variables needed for post-processing later
EddyProc.C <- sEddyProc$new('DE-Tha', EddyDataWithPosix.F,
    c('NEE','Rg','Tair','VPD', 'Ustar'))
```

## B2   Estimating the uStar threshold distribution

The second step is the estimation of the distribution of uStar thresholds to identify periods of low friction velocity (uStar), where
NEE is biased low. Discarding periods with low uStar is one of the largest sources of uncertainty in aggregated fluxes. Hence,
several quantiles of the distribution of the uncertain uStar threshold are estimated by a bootstrap.

5   The friction velocity, uStar, needs to be in a column named "Ustar" of the input dataset.

```
uStarTh <- EddyProc.C$sEstUstarThresholdDistribution(
  nSample = 100L, probs = c(0.05, 0.5, 0.95))
#filter(uStarTh, aggregationMode == "year")
select(uStarTh, -seasonYear)
```

```
##   aggregationMode  season    uStar        5%        50%        95%
## 1          single    <NA> 0.4162500 0.3658125 0.4414583 0.6197520
## 2            year    <NA> 0.4162500 0.3658125 0.4414583 0.6197520
## 3          season 1998001 0.4162500 0.3658125 0.4414583 0.6197520
## 4          season 1998003 0.4162500 0.3198846 0.4037981 0.5615500
## 5          season 1998006 0.3520000 0.3192476 0.3858571 0.4393708
## 6          season 1998009 0.3369231 0.2195485 0.3906458 0.5235175
## 7          season 1998012 0.1740000 0.1829667 0.4189286 0.6180982
```

The output reports uStar estimates of 0.42 for the orignal data and 0.37, 0.44, 0.62 for lower, median, and upper quantile
15  of the estimated distribution. The threshold can vary between periods of different surface roughness, e.g. before and after
harvest. Therefore, there are estimates for different time periods of the year, called seasons, reported as different rows. These
season-estimates can be aggregated to entire years or to a single value across years, reported by rows with corresponding
aggregation mode.

The subsequent post processing steps will be repeated using the three quantiles of the uStar distribution. They require to specify
20  a uStar-threshold for each season and a suffix to distinguish the outputs related to different thresholds.





For this example of an evergreen forest site, the same annually aggregated uStar threshold estimate will be chosen for each of the seasons within a year. In order to distinguish the automatically generated columns, the column names of the estimation results are written to variable `uStarSuffixes`.

```
uStarThAnnual <- usGetAnnualSeasonUStarMap(uStarTh)[-2]
uStarSuffixes <- colnames(uStarThAnnual)[-1]
print(uStarThAnnual)
```

```
##    season       U05       U50      U95
## 1 1998001 0.3658125 0.4414583 0.619752
## 2 1998003 0.3658125 0.4414583 0.619752
## 3 1998006 0.3658125 0.4414583 0.619752
## 4 1998009 0.3658125 0.4414583 0.619752
## 5 1998012 0.3658125 0.4414583 0.619752
```

## B3 Gap-filling

The second post-processing step is filling the gaps using information of the valid data. In this case, the same annual uStar threshold estimate is used for each season, as described above, and the uncertainty will be computed also for valid records (`FillAll`).

```
EddyProc.C$sMDSGapFillAfterUStarDistr('NEE',
   UstarThres.df = uStarThAnnual,
   UstarSuffix.V.s = uStarSuffixes,
     FillAll = TRUE
)
```

The screen output (not shown here) already shows that the uStar-filtering and gap-filling was repeated for each given estimate of the uStar threshold , i.e. column in `uStarThAnnual`, with marking 22% to 38% of the data as gap.

For each of the different uStar threshold estimates, a separate set of output columns of filled NEE and its uncertainty is generated, distinguished by the suffixes given with `uStarSuffixes`. Suffix "_f" denotes the filled value and "_fsd" the estimated standard devation of its uncertainty.

```
grep("NEE_.*_f$",names(EddyProc.C$sExportResults()), value = TRUE)
grep("NEE_.*_fsd$",names(EddyProc.C$sExportResults()), value = TRUE)
```

```
## [1] "NEE_U05_f" "NEE_U50_f" "NEE_U95_f"
## [1] "NEE_U05_fsd" "NEE_U50_fsd" "NEE_U95_fsd"
```





## B4    Partitioning net flux into GPP and R$_{eco}$

The third post-processing step is partitioning the net flux (NEE) into its gross components GPP and R$_{eco}$. The partitioning
algorithm needs a precise criteria between night-time and day-time. Therefore, the specification of geographical coordinates and
time zone need to be provided to allow computing exact solar time of sunrise and sunset. Further, the missing values in the used
5    meteorological data need to be filled.

```
EddyProc.C$sSetLocationInfo(Lat_deg.n = 51.0, Long_deg.n = 13.6, TimeZone_h.n = 1)
EddyProc.C$sMDSGapFill('Tair', FillAll.b = FALSE)
EddyProc.C$sMDSGapFill('VPD', FillAll.b = FALSE)
```

Now we are ready to invoke the partitioning, here by the night-time approach, for each of the several filled NEE columns.

```
#variable uStarSuffixes was defined above at the end of uStar threshold estimation
resPart <- lapply(uStarSuffixes, function(suffix){
                    EddyProc.C$sMRFluxPartition(Suffix.s = suffix)
              })
```

The results are stored in columns `Reco` and `GPP_f` modified by the respective uStar threshold suffix.

```
grep("GPP.*_f$|Reco",names(EddyProc.C$sExportResults()), value = TRUE)
```

```
## [1] "Reco_U05"  "GPP_U05_f" "Reco_U50"  "GPP_U50_f" "Reco_U95"  "GPP_U95_f"
```

The visualizations of the results by a fingerprint plot gives a compact overview.

```
EddyProc.C$sPlotFingerprintY('GPP_U50_f', Year.i = 1998)
```

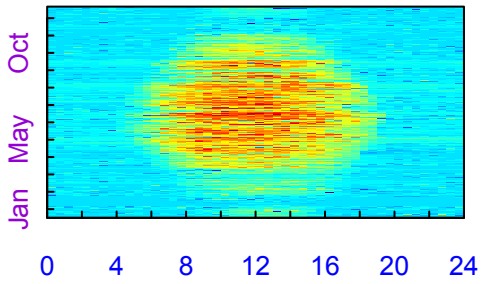

## B5    Estimating the uncertainty of aggregated results

First, the mean of the GPP across all the year is computed for each uStar-scenario and converted from $\mu mol\, CO_2\, m^{-2} s^{-1}$ to
$gC\, m^{-2} yr^{-1}$.




```r
FilledEddyData.F <- EddyProc.C$sExportResults()
#suffix <- uStarSuffixes[2]
GPPAggCO2 <- sapply( uStarSuffixes, function(suffix) {
    GPPHalfHour <- FilledEddyData.F[[paste0("GPP_",suffix,"_f")]]
    mean(GPPHalfHour, na.rm = TRUE)
})
molarMass <- 12.011
GPPAgg <- GPPAggCO2 * 1e-6 * molarMass * 3600*24*365.25
print(GPPAgg)
```

```
##      U05      U50      U95
## 1894.097 1956.090 1985.817
```

The difference between those aggregated values is a first estimate of uncertainty range in GPP due to uncertainty of the uStar threshold.

```r
(max(GPPAgg) - min(GPPAgg)) / median(GPPAgg)
```

5    In this run of the example a relative error of about 4.7% is inferred.

For a better but more time consuming uncertainty estimate, specify a larger sample of uStar threshold values, for each repeat the post-processing, and compute statistics from the larger sample of resulting GPP columns. This can be achieved by specifying a larger sequence of quantiles when calling `sEstUstarThresholdDistribution`.

```r
sEstUstarThresholdDistribution(
  nSample = 200, probs = seq(0.025,0.975,length.out = 39) )
```

**B6    Storing the results in a csv-file**

10    The results still reside inside the `sEddyProc` class. To export them to an R Data.frame, the newly generated columns need to be appended to the columns with the original input data. Then this data.frame is written to a text file in a temporary directory.

```r
FilledEddyData.F <- EddyProc.C$sExportResults()
CombinedData.F <- cbind(EddyData.F, FilledEddyData.F)
fWriteDataframeToFile(CombinedData.F, 'DE-Tha-Results.txt', Dir.s = tempdir())
```

**B7    Specifying seasons where uStar thresholds differs**

With changing surface roughness, e.g. on harvest or leaf-fall, also the uStar-NEE relationship can change. Therefore the uStar threshold needs to be re-estimated at different times of the year, called seasons. The default uses continuous sea-



sons, for details see section 3.2.1. In order to yield results corresponding to DP06, the user can specify `seasonFactor.v = usCreateSeasonFactorMonthWithinYear( EddyData.C$sDATA$sDateTime, startMonth= c(3,6,9,12))` as an argument to routine `sEstUstarThreshold`. By default the annual aggregate of the season thresholds, i.e. maximum across seasons, is used to identify unfavorable conditions. The seasonal estimates can also be used instead.

Moreover, the users can specify also other user-defined seasons, e.g. when harvest dates are known (see package vignette DEGebExample). They can create a grouping by specifying exact starting days of the periods by function `usCreateSeasonFactorYdayYear`, or they can provide a column with the data that indicates e.g. the same group for two wet seasons. Each season is associated to the year corresponding to the center day between first and last day of the season.

With all methods, there is a required minimum number of 160 records within a season. If there are too few records, the data of
the seasons within one year are combined and the uStar threshold for this seasons is set to the estimate obtained for the data of the entire year.

## Appendix C: Additional benchmark statistics

This section provides figures, in addition to the main figures presented in the benchmark section (3), that require the reader being comfortable with histograms and probability distribution functions.

The histograms display how often a certain value occurs across all the site-months. A centering of a difference away from zero or for a ratio away from one denotes a bias.

### C1   uStar Threshold estimation

There was a bias in uStar estimation (Fig. C1). However, the bias and the differences were only a small fraction of the uncertainty of the estimate itself. Moreover, the bias did not propagate through estimated NEE based on different uStar values (Fig. C2).

There was a significant difference in the estimate of uncertainty of the uStar thresholds (Fig. C3). `REddyProc` only estimated half of the uncertainty due to acknowledging the seasons of similar conditions also during bootstrap. This lower uncertainty propagated to the uncertainty estimates of NEE (Fig. C4)

### C2   Gap-filling

The evaluation of annual aggregated NEE data obtained with `REddyProc` and the BGC online tool showed good agreement
across sites.

Note, that the aggregated NEE data contained both, measured and gap-filled data for the purpose of evaluating the impact of the processing on the aggregated NEE. The determination coefficient ($R^2$) showed a very good agreement between the two





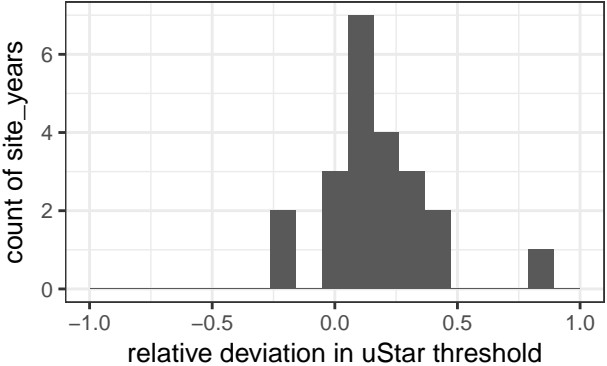

**Figure C1.** Histogram of differences between uStar threshold estimates of different methods (DP06 - `REddyProc`) normalized by the uncertainty range (90% quantile - 10% quantile estimated by DP06). Absolute values are smaller than one meaning that the difference between the methods is smaller than range of estimates by the DP06 method only. The clustering of values on the positive side suggests a bias towards larger values with DP06.

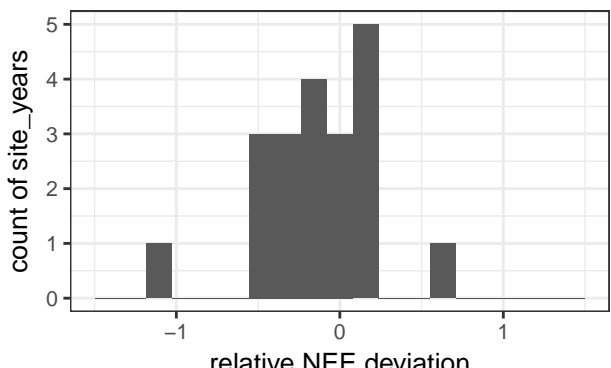

**Figure C2.** Histogram of differences between annual NEE based on uStar estimates of different methods (Papale - `REddyProc`) normalized by the uncertainty range of NEE due to uStar (NEE based on 90% uStar quantile - NEE based on 10% uStar quantile estimated by DP06). Absolute values are mostly smaller than one, meaning that the difference in annual NEE between methods is smaller than the uncertainty due to uncertainty of uStar threshold from DP06 only.

methods both at annual and monthly time scale (Table C1). The relative mean absolute error (RMAE) is low: about -3 % for annual aggregation and -0.47 % for monthly aggregation. Both Modeling Efficiency (EF) and Mean Bias Error (MBE) showed a very low bias between the two products for both monthly and annual aggregations (Table C1).





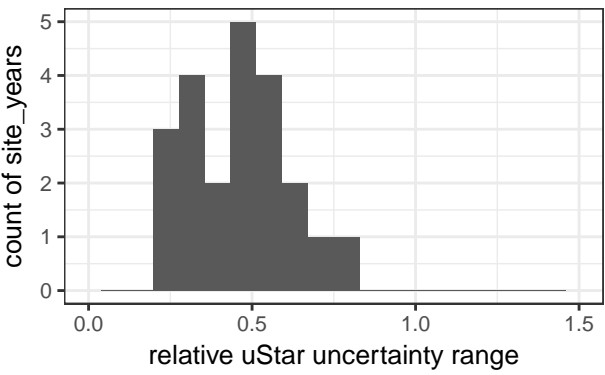

**Figure C3.** Histogram of ratio (`REddyProc` / DP06) of uncertainty ranges of uStar (90% quantile - 10% quantile). The clustering around 0.5 shows that the estimated uncertainty with `REddyProc` is only about half the uncertainty estimated by DP06.

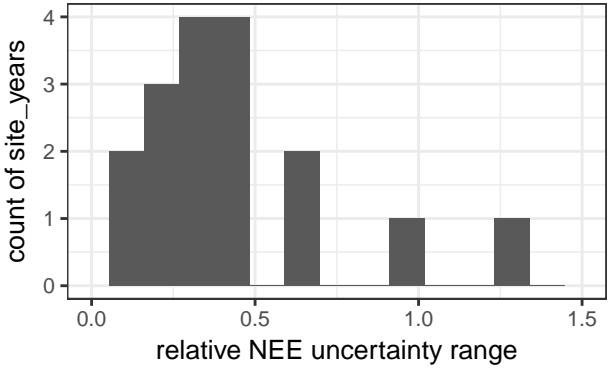

**Figure C4.** Histogram of ratio (`REddyProc` / DP06) of uncertainty ranges of annual NEE (NEE based on 90% quantile of uStar threshold - NEE based on 10% qunatile of uStar threshold). The clustering below a value of one indicates that the lower uncertainty in uStar threshold with `REddyProc` also propagates to lower uncertainty in annual NEE.





**Table C1.** Gap-filling evaluation statistics of yearly (Fig. 6) and monthly cumulated and gap-filled NEE data obtained with `REddyProc` and the BGC online tool. Statistic abbreviations are explained in section C2.

|         | Yearly | Monthly |
|--------:|-------:|--------:|
| N       | 25     | 281     |
| pearson | 0.99   | 1.00    |
| MBE     | -0.02  | 0.00    |
| RMBE    | 2.8%   | -0.08%  |
| MAE     | 0.02   | 0.00    |
| RMAE    | -3.0%  | -0.5%   |
| RMSE    | 0.09   | 0.01    |
| R2      | 0.98   | 1.00    |
| EF      | 0.98   | 1.00    |





## C3 Night-time partitioning

The results show good agreement between $R_{eco}$ estimated using `REddyProc` and the BGC online tool for R2 and EF (Table C2). The relative RMSE is 6.56 and 13.45 % for yearly and monthly aggregation, respectively. The high RRMSE is due to few site years as reported in Table C3, and this is confirmed by the lower RMAE (3.87 % and 6.33 % for yearly and monthly, respectively), which is less sensitive to outliers.

The difference in $R_{eco}$ related to the selection of night-time data are not negligible: the differences in RRMSE of 1.03% and negligible differences in $R2$. Also, the use of $E_0$ prescribed from the BGC online tool lead to negligible difference in R2 of about 0.02. Therefore, though very small, the selection of night-time data is the most important difference introduced by `REddyProc`.

**Table C2.** Nighttime partitioning evaluation statistics across sites of annually (Fig. 7) and monthly aggregated ecosystem respiration ($R_{eco}$) estimated with `REddyProc` and the BGC online tool.

|  | Yearly | Monthly |
|---|---|---|
| N | 25 | 297 |
| pearson | 0.99 | 0.99 |
| MBE | 25.4 | 2.15 |
| RMBE | 2.2% | 2.2% |
| MAE | 44.8 | 6.17 |
| RMAE | 3.8% | 6.3% |
| RMSE | 75.6 | 13.2 |
| RRMSE | 6.5% | 13.5% |
| R2 | 0.99 | 0.98 |
| EF | 0.99 | 0.97 |

## C4 Day-time partitioning

The time-variable estimate of temperature sensitivity of ecosystem respiration with day-time partitioning is a significant source of uncertainty for gross fluxes GPP and $R_{eco}$. `REddyProc` accounts for this previously unaccounted uncertainty for estimating uncertainty of these gross fluxes by a bootstrap (Fig. C5). The estimate of annual uncertainty in Fig. C5 is a low estimate, because it assumed no correlation between half-hourly errors. An improved quantification of correlations requires the full variance-covariance matrix of the LRC parameter fits (Lasslop et al., 2010; Menzer et al., 2013), which were not available for the BGC online tool.

The introduced uncertainty is reduced by smoothing the $E_0$ estimates across several successive windows (Fig. C5 top) before estimating parameters of the LRC. This smoothing has also an effect on predicted half-hourly gross fluxes (Fig. C5 bottom).





|  | RMBE | RMAE | RRMSE | R2 | EF |
|---|---|---|---|---|---|
| CA-NS7 | 15.12 | 16.29 | 32.64 | 0.96 | 0.86 |
| CA-TP3 | -1.99 | 25.10 | 38.59 | 0.87 | 0.82 |
| DE-Hai | 6.00 | 6.48 | 10.40 | 0.99 | 0.96 |
| DE-Tha | 2.83 | 3.52 | 6.19 | 0.99 | 0.99 |
| DK-Sor | 3.50 | 4.45 | 7.34 | 1.00 | 0.99 |
| ES-ES1 | 1.68 | 3.34 | 4.59 | 0.93 | 0.92 |
| ES-VDA | 3.96 | 9.98 | 15.93 | 0.95 | 0.93 |
| FI-Hyy | 2.72 | 3.05 | 4.47 | 1.00 | 1.00 |
| FI-Kaa | -4.52 | 5.51 | 9.07 | 0.99 | 0.99 |
| FR-Gri | 3.96 | 4.14 | 6.35 | 1.00 | 0.99 |
| FR-Hes | 4.11 | 5.06 | 9.06 | 0.99 | 0.99 |
| FR-Lq1 | -2.89 | 5.76 | 7.24 | 0.99 | 0.99 |
| FR-Lq2 | -8.27 | 9.49 | 11.49 | 0.99 | 0.97 |
| FR-Pue | 0.65 | 1.27 | 1.55 | 1.00 | 1.00 |
| IE-Dri | -0.03 | 2.41 | 3.24 | 1.00 | 1.00 |
| IL-Yat | 1.66 | 2.89 | 3.72 | 0.99 | 0.99 |
| IT-Amp | 24.34 | 29.38 | 45.33 | 0.91 | 0.48 |
| IT-MBo | 2.03 | 3.00 | 4.49 | 1.00 | 1.00 |
| IT-SRo | -1.46 | 1.86 | 2.10 | 1.00 | 0.99 |
| PT-Esp | 9.06 | 12.08 | 19.94 | 0.62 | 0.45 |
| RU-Cok | -0.76 | 28.54 | 38.98 | 0.27 | -0.07 |
| SE-Nor | -1.05 | 2.09 | 3.62 | 1.00 | 1.00 |
| US-Ton | 2.93 | 7.45 | 11.82 | 0.93 | 0.92 |
| VU-Coc | 0.73 | 2.64 | 3.83 | 0.93 | 0.93 |

**Table C3.** Nighttime partitioning evaluation statistics at sites level of the of ecosystem respiration ($R_{eco}$) estimated with `REddyProc` and the BGC online tool.

Results of daytime partitioning are sensitive to subtle details of the procedure. Hence, there is quite much scatter introduced by the differences of processing `REddyProc` with default options or with options that maximize compatibility with the BGC online tool (Fig. C7 top). One such a subtle options is to decrease or to not account for the unreasonably high leverage of some observations during the fit to the light-response curve by some records having a very small estimate of its uncertainty (Fig. C7 bottom). Due to the sensitivities of the day-time partitioning, there are still differences between `REddyProc` with compatibility options and the BGC online tool (Fig. C8).





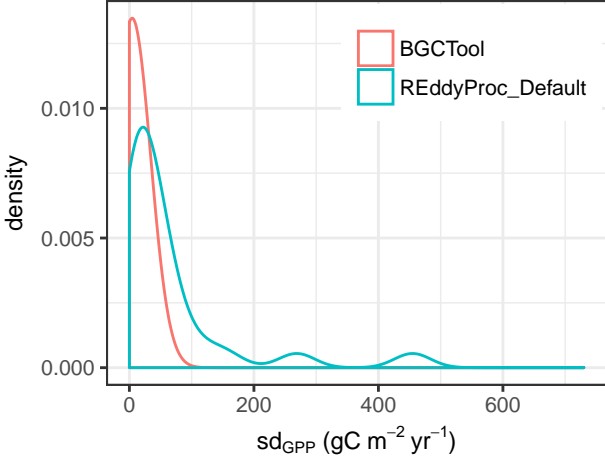

**Figure C5.** Density plot of estimated standard deviation of uncertainty of the annually aggregated GPP across sites due to uncertainty in parameters estimation during day-time based flux partitioning. Higher estimates with `REddyProc` are caused by taking into account the uncertainty in temperature sensitivity, $E_0$.

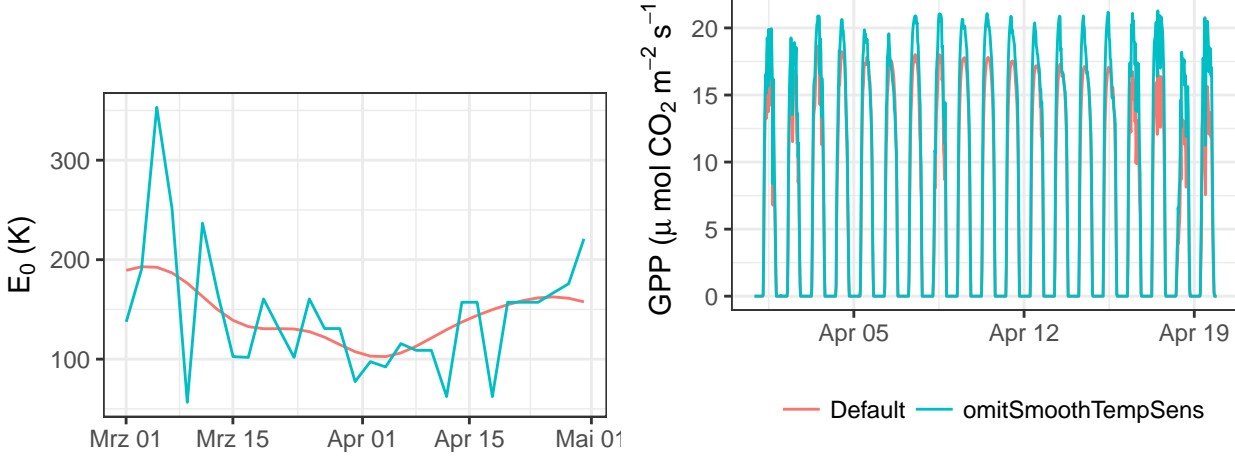

**Figure C6.** Effects of smoothing (top) successive estimates of temperature sensitivity, $E_0$, on predicted GPP (bottom) for site PT-Esp.

*Acknowledgements.* This work used eddy covariance data acquired and shared by the FLUXNET community, including these networks: AmeriFlux, AfriFlux, AsiaFlux, CarboAfrica, CarboEuropeIP, CarboItaly, CarboMont, ChinaFlux, Fluxnet-Canada, GreenGrass, ICOS, KoFlux, LBA, NECC, OzFlux-TERN, TCOS-Siberia and USCCC. The ERA-Interim reanalysis data are provided by ECMWF and processed by LSCE. The FLUXNET eddy covariance data processing and harmonization was carried out by the European Fluxes Database Cluster, AmeriFlux Management Project, and Fluxdata project of FLUXNET, with the support of CDIAC and ICOS Ecosystem Thematic Center, and the OzFlux, ChinaFlux and AsiaFlux offices.



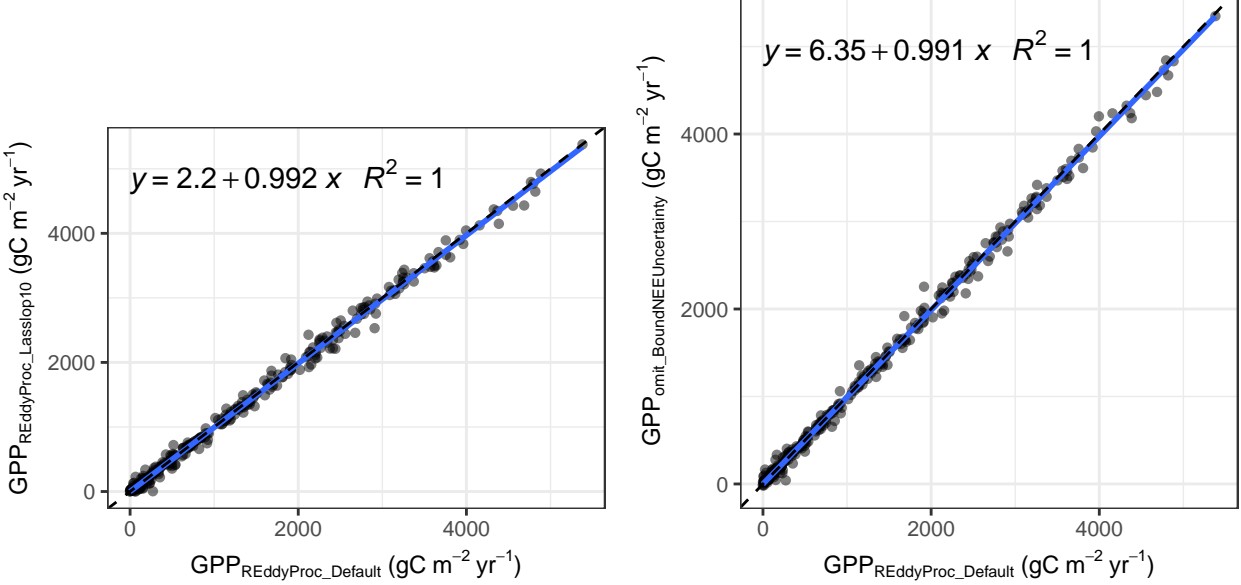

**Figure C7.** Sensitivity of estimated monthly GPP fluxes to specific processing details results in scatter between GPP predictions based on different `REddyProc` options. Most of the differences between default options and compatibility options (top) are caused by differences in weighting different records during the fit (bottom.)

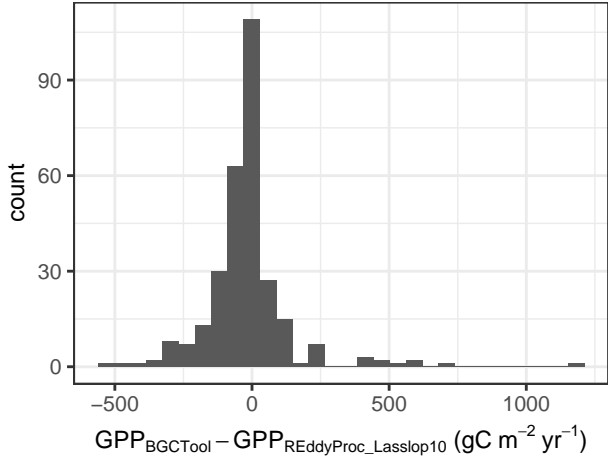

**Figure C8.** Histogram of difference between monthly GPP predictions of `REddyProc` with Lasslop10 compatibility options and the BGC online tool.

The authors acknowledge Dario Papale, Gilberto Pastorello and Trevor F Keenan for the discussions on the benchmarking of REddyProc and PvWave code. M. M. and M. R. acknowledge the Alexander Von Humbold foundation that funded part of this research activity with the Max Planck Research Preis to Markus Reichstein. M. M. acknowledge the MSCA-ITN project TRUSTEE.





L.Š. was supported by the Ministry of Education, Youth and Sports of the Czech Republic within the CzeCOS program, grant number
LM2015061, and within the National Sustainability Program I (NPU I), grant number LO1415.



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
