# Peer review of "Basic and extensible post-processing of eddy covariance flux data with REddyProc."

_Biogeosciences, 2018_

## Referee Comment (RC1) · Anonymous Referee #1 · 7 Mar 2018

The paper adds to a growing segment of extensive method descriptions for reproducible computational research. While the technical focus is laudable, it is also the reason that the manuscript in its current form somewhat misses the scope of BG ("interactions between the biological, chemical, and physical processes"; https://www.biogeosciences.net/index.html). I imagine this might be part of the reason that the other referees declined. Alternative Copernicus journals like AMT ("techniques of data processing"; https://www.atmospheric-measurement-techniques.net/index.html) or GMD ("statistical models", "technical papers"; https://www.geoscientific-model-development.net/) should provide a much better fit.

Here a few points for consideration in such re-submission: - I suggest shortening the

manuscript. It should be straightforward to consolidate 22 pages of heavy methodological detail by ∼1/3, and focus on novel aspects.

- l 5: "standard tools available in open source environment for processing high-frequency (10 or 20 Hz) data into half-hourly quality checked fluxes". At this time open-source environments for eddy-covariance data processing that actually facilitate open development are only emergent. REddyProc provides a substantial and much appreciated contribution to this movement. I suggest to either substantiate the claim of an abundance of open-source high-frequency data processing environments through providing examples, or to provide a more differentiated overview.

- l. 8: While it is true that R is a cross-platform language, this does not mean that research is reproducible by using an R-package across platforms. Known as "dependency hell", installing e.g. REddyProc on a standard Debian Linux distribution requires the co-installation of several operating-system-side libraries (libudunits2-0, libudunits2-dev, udunits-bin, libnetcdf-dev) and even more R-side dependencies (backports, praise, evaluate, highr, mlegp, logitnorm, ncdf4, RNetCDF, minpack.lm, segmented, rprojroot, testthat, knitr). In some operating systems such as Windows, there is hardly any automation available for resolving operating-system-side dependencies, making R-packages with heavy dependencies inaccessible to less experienced users. Most importantly, dependency resolution itself is not reproducible among operating systems, thus rendering reproducible research impossible. A balanced discussion of how REddyProc can be used for reproducible research alongside examples for dependency resolution would add much substance and usability to the manuscript.

- l 15: It could be pointed out that REddyProc has already been adopted for computational research by the flux community, such as in Metzger et al. (2017). These authors also point to a community solution for "dependency hell", an pre-compile REddyProc alongside its dependencies into compute images that contain a turn-key, reproducible and shareable processing environment.

[Figure]

References Metzger, S., Durden, D., Sturtevant, C., Luo, H., Pingintha-Durden, N., Sachs, T., Serafimovich, A., Hartmann, J., Li, J., Xu, K., and Desai, A. R.: eddy4R 0.2.0: a DevOps model for community-extensible processing and analysis of eddy-covariance data based on R, Git, Docker, and HDF5, Geosci. Model Dev., 10, 3189-3206, doi:10.5194/gmd-10-3189-2017, 2017.

---

## Short Comment (SC1) · 7 Mar 2018

We thank referee #1 for his constructive comments.

The main focus of the manuscript is the method comparison to existing tools. Most technical details have been moved to appendices, that we want to keep in order to support the readers in their applications. We will experiment with moving more detail to appendices and shortening or moving current appendices to online supplements. However, most of the left detail is required to understand the differences in results between methods. Moreover, the manuscript should serve as a reference for the methods and therefore we strive to state the details.

The length of the manuscript results also from the fact, that essentially three different

issues are explored, and all of them are compared to existing methods: 1) filtering, i.e. uStar threshold estimation 2) gap-filling, and 3) flux-partitioning. When splitting the manuscript into three papers, big parts, e.g. in the introduction, will be be redundant.

While the suggested alternative journals maybe fit the content slightly better, we want to target an audience of researchers who use EC flux data in their studies. We argue that Biogeosciences is the best open access journal to reach this audience.

We will tackle the issue of "dependency hell" by providing a Docker image that already includes all required system libraries and R-packages.

─────────────────────────────

---

## Referee Comment (RC2) · Anonymous Referee #2 · 11 Mar 2018

**1    General comments**

The paper describes the R package REddyProc containing tools for post-processing of eddy covariance data with a focus on $CO_2$. The REddyProc package contains tools for reading halfhourly data from different formats, estimating $u_*$ thresholds, gapfilling, flux-partioning, data visualisation, and estimation of uncertainties.

The results obtained with the routines of the package are compared with other state-of-the art tools. This resulted in no significant differences in the results from the different tools on a montly or annual scale. The REddyProc package contain a more sophisticated way of the treatment of seasons and the possiblity to run and take advantage of multi-year data sets.

[Figure]

The paper provides a good description of the rationale of the package and the diffence in calculation method between this package and earlier available tools. The paper also includes an appendix with example of how to run the routines on a full annual dataset.

In general, I find that the paper is well written and documented, and that it can provide a very useful reference for scientist using the package. I have some suggestions for improvements in the specific comment below. Providing this package is very good for the flux reasearch community and by using standardized methods the results from different research groups become much more comparable.

**2 Specific comments**

Table 1. I find it a bit lazy to write "uStar" in stead of the correct symbol "$u_*$".

Figure 2. Caption: I suppose it should be "night-time NEE". The "season–temperature subset" is only understandable after careful reading of the main text.

p.6, l.12-13: It is not completely clear from the text which time-frame is used for the flagging. What is the chosen aggregation period for a "robust $u_*$ estimate" (l. 10).

p.7, l.8.: Probably this is the first time in the paper the "BGC online tool" is mentioned, and should thus be explained and referenced here.

p.9, l.22: The abbreviation "LRC" should be explained.

p.12, l.13: The choice of the software used for comparison (here called "benchmarking") should be explained better. The $u_*$ filtering is compared to Papale et al. (2006) and the gap-filling and flux partioning to the BGC online tool. Do these represent (previous) state-of-the-art or have exactly those routines been used for calculating the FLUXNET data-sets?

p.14, l.6: Should read "To explore these consequences ..."

Fig.4+5: I find it quite interesting that there seems to be a fairly large scatter in the estimated $u_*$ thresholds, but this does not seem to translate into a similar scatter in the annual NEE estimates. It would be valuable to have a further discussion on this. Does it e.g. mean that the value of the $u_*$ threshold is not very important? A lower $u_*$ threshold means that more data are kept which could potentially lower the uncertainty on the annual estimate.

Fig. 6: The symbols of the quality flags are very difficult to read and distinguish. The legend refers to the sub-figures as "top" and "bottom". Here it should be "left" and "right".

Appendix B: I tried out the REddyProc package on my MacBook, First I tried to update all packages and after a little tweaking I managed to load REddyProc. Following the example went very smooth until running the function sEstUstarThresholdDistribution() where I got the message "Error: could not find function "sEstUstarThresholdDistribution". This is probably a minor trivial issue in my implementation that can be solved.

---

## Author Comment (AC1) · 4 Apr 2018

This is an extended version of the immediate short comment by Thomas Wutzler, agreed by all coauthors. Here we repeat each reviewer's comment (RC) before each of our author's comment (AC) replies in blue.

We thank referee 1 for his constructive comments.

**RC**: The paper adds to a growing segment of extensive method descriptions for reproducible computational research. While the technical focus is laudable, it is also the reason that the manuscript in its current form somewhat misses the scope of BG ("interactions between the biological, chemical, and physical processes"; https://www.biogeosciences.net/index.html). I imagine this might

be part of the reason that the other referees declined. Alternative Copernicus journals like AMT ("techniques of data processing"; https://www.atmospheric-measurement-techniques.net/index.html) or GMD ("statistical models", "technical papers"; https://www.geoscientific-model-development.net/) should provide a much better fit.

**AC**: While the suggested alternative journals maybe fit the content slightly better, they target an audience primarily interested in data processing or model development. Contrary, we want to target an audience of researchers who *use* EC flux data in their studies. We argue that Biogeosciences is the best open access journal to reach this audience. For instance Biogeosciences Journal recently hosted a special issue on eddy covariance data collected in the Australia and New Zealand, which include also more methodological papers "OzFlux: a network for the study of ecosystem carbon and water dynamics across Australia and New Zealand, 2016". Also a search of the keyword "eddy covariance" in the main text on April 4th, 2018 give back 1620 records. We also looked at the number of views in the open review phase and they were more than 350 (April 4th), supporting the idea that Biogeosciences can be the right audience argument for the article.

**RC**: Here a few points for consideration in such re-submission: - I suggest shortening the Discussion paper manuscript. It should be straightforward to consolidate 22 pages of heavy methodological detail by about 1/3, and focus on novel aspects.

**AC**: We agree with the reviewer that the manuscript can be shortened and in the revised version we will work in this direction. In particular we will move some paragraphs (sec 2.2.3, Fig3, parts of sec 2.3.3 )) in the appendix and part of the appendices to online supplementary materials (Appendix A1 and C). We strive to report the details in some way for reproducibility.

The length of the manuscript results also from the fact, that essentially three different issues are explored, and all of them are compared to existing methods: 1) filtering,
i.e. $u_*$ threshold estimation 2) gap-filling, and 3) flux-partitioning. When splitting the manuscript into three papers, big parts, e.g. in the introduction, will be redundant.

**RC**: - l 5: "standard tools available in open source environment for processing high frequency (10 or 20 Hz) data into half-hourly quality checked fluxes". At this time open-source environments for eddy-covariance data processing that actually facilitate open development are only emergent. REddyProc provides a substantial and much appreciated contribution to this movement. I suggest to either substantiate the claim of an abundance of open-source high-frequency data processing environments through providing examples, or to provide a more differentiated overview.

**AC**: We agree with the reviewer that open source tools for eddy covariance processing are only now emerging. There are now available some open source tools for processing of high frequency data, for instance the Eddy Pro software, which is open source and widely used. Moreover, there are also an increasing amount of packages (see this URL[1]): We will modify the sentence to clarify ths aspect.

**RC**: - l. 8: While it is true that R is a cross-platform language, this does not mean that research is reproducible by using an R-package across platforms. Known as "dependency hell", installing e.g. REddyProc on a standard Debian Linux distribution requires the co-installation of several operating-system-side libraries (libudunits2-0, libudunits2-dev, udunits-bin, libnetcdf-dev) and even more R-side dependencies (backports, praise, evaluate, highr, mlegp, logitnorm, ncdf4, RNetCDF, minpack.lm, segmented, rprojroot, testthat, knitr). In some operating systems such as Windows, there is hardly any automation available for resolving operating-system-side dependencies, making R-packages with heavy dependencies inaccessible to less experienced users. Most importantly, dependency resolution itself is not reproducible among operating systems, thus rendering reproducible research impossible. A balanced discussion of how REddyProc can be used for reproducible research alongside examples for dependency
* * *
[1]http://fluxnet.fluxdata.org/2017/10/10/toolbox-a-rolling-list-of-softwarepackages-for-flux-related-data-processing/

resolution would add much substance and usability to the manuscript.

**AC**: The library dependency issue are caused by the NetCDF packages. We listed this packages as "suggests" instead of "depends". Hence, REddyProc can be used and installed without these dependencies really with a single line as exemplified on page 22 line 14 (We tested it also on a standard DEBIAN distribution using docker). Only when trying to read NetCDF files, REddyProc issues an error advising to install these packages before.
For users who want to read NetCDF files but have not yet installed the required sysem libraries, together with the revised manuscript, we will provide a Docker image that already includes all required system libraries and R-packages.
However, in the revised manuscript we wil keep these technical details short for the sake manuscript length and targeted audience.

**RC**: - l 15: It could be pointed out that REddyProc has already been adopted for computational research by the flux community, such as in Metzger et al. (2017). These authors also point to a community solution for "dependency hell", an pre-compile REddyProc alongside its dependencies into compute images that contain a turn-key, reproducible and shareable processing environment.

**AC**: Thanks for this suggestion. We will cite the paper when referring to the provided Docker image, but will keep technical details short because of the targeted audience.

---

## Author Comment (AC2) · 4 Apr 2018

**RC**: The paper describes the R package REddyProc containing tools for post-processing of eddy covariance data with a focus on $CO_2$. The REddyProc package contains tools for reading halfhourly data from different formats, estimating u* thresholds, gapfilling, flux-partioning, data visualisation, and estimation of uncertainties. The results obtained with the routines of the package are compared with other state-of-the art tools. This resulted in no significant differences in the results from the different tools on a montly or annual scale. The REddyProc package contain a more sophisti cated way of the treatment of seasons and the possiblity to run and take advantage of multi-year data sets. Discussion paperThe paper provides a good description of the rationale of the package and the diffence in calculation method between this package

and earlier available tools. The paper also includes an appendix with example of how to run the routines on a full annual dataset. In general, I find that the paper is well written and documented, and that it can provide a very useful reference for scientist using the package. I have some suggestions for improvements in the specific comment below. Providing this package is very good for the flux reasearch community and by using standardized methods the results from different research groups become much more comparable.

**AC**: We thank reviewer 2 for his encouraging comments.

**0.1 Specific comments**

**RC**: Table 1. I find it a bit lazy to write "uStar" instead of the correct symbol "$u_*$".

**AC**: We want to keep consistency throughout the manuscript. Since in the programming source code and most processing output the multiplication character cannot be used, we consistently used "uStar". If required by the editor, we sacrifice a bit of consistency in favor of convention and we will change the manuscript to use the symbol "$u_*$" in places other than source code and processing output in the revised version.

**RC**: Figure 2. Caption: I suppose it should be "night-time NEE". The "season–temperature subset" is only understandable after careful reading of the main text.

**AC**: In a revised version we will adapt the figure caption 2 to: "Concept of the $u_*$-filter: Night-time NEE at low $u_*$ friction velocities below a threshold, i.e with low turbulence, is biased low compared higher $u_*$ with otherwise similar environmental conditions. The $u_*$ threshold (dashed line) is estimated by a moving point method on $u_*$ bins (crosses) across half-hourly records (circles) here for a subset of data from DE-Tha."

**RC**: p.6, l.12-13: It is not completely clear from the text which time-frame is used for the flagging. What is the chosen aggregation period for a "robust u* estimate" (l. 10).

**AC**: $u_*$ values of each half hours are compared to the threshold determined for the corresponding season. In addition each half-hour with too low $u_*$ in the previous half-hour is flagged to be invalid. We will describe this better in a revised version.
The aggregation periods, i.e. seasons, are specified by the user and default to three months, starting in months, 3,6,9, and 12, as described at page 14 lines, 14ff.

**RC**: p.7, l.8.: Probably this is the first time in the paper the "BGC online tool" is mentioned, and should thus be explained and referenced here.

**AC**: Thanks for noting. In a revised version, we will keep the description of the BGC online tool in the benchmarking section, but insert a link in this method description section.

**RC**: p.9, l.22: The abbreviation "LRC" should be explained.

**AC**: Thanks for noting. In a revised version, we will keep the description of the Light response curve (LRC) in the specific day-time partitioning section, but write in this general section "fits a model to observations of daytime NEE and global radiation"

**RC**: p.12, l.13: The choice of the software used for comparison (here called "benchmarking") should be explained better. The u* filtering is compared to Papale et al. (2006) and the gap-filling and flux partitioning to the BGC online tool. Do these represent (previous) state-of-the-art or have exactly those routines been used for calculating the FLUXNET data-sets?

**AC**: The required more detailed description of the role of the benchmark tools is currently described in the more specific sections, 3.2 and 3.5. In a revised version, we want to keep the details in the specific sections, but will move some parts from section 3.5 to section 3.3.

**RC**: p.14, l.6: Should read "To explore these consequences ..."

**AC**: Thanks for noting the typo.

**RC**: Discussion paperFig.4+5: I find it quite interesting that there seems to be a fairly large scatter in the estimated u* thresholds, but this does not seem to translate into a similar scatter in the annual NEE estimates. It would be valuable to have a further discussion on this. Does it e.g. mean that the value of the u* threshold is not very important? A lower u* threshold means that more data are kept which could potentially lower the uncertainty on the annual estimate.

**AC**: We also did not expect to which extent the scatter in u* thresholds did not propagate to the aggregated flux results, and discussed this at sections 3.2.3: "indicates that the sensitivity of NEE to the $u_*$ threshold estimate in the inferred ranges is low, which also explains the large uncertainty of the $u_*$ threshold estimate". In a revised version we will experiment with elaborating a bit more on it, while keeping in mind the manuscript-length constraints.

**RC**: Fig. 6: The symbols of the quality flags are very difficult to read and distinguish. The legend refers to the sub-figures as "top" and "bottom". Here it should be "left" and "right".

**AC**: In the final (as opposed to review) two column mode formatting, the labels "top" and "bottom" are the correct ones. The symbol transparency is a compromise between reading a single symbol and conveying the message that most of the values overplot at the 1:1 line. In a revised version we will adjust towards less transparency.

**RC**: Appendix B: I tried out the REddyProc package on my MacBook, First I tried to update all packages and after a little tweaking I managed to load REddyProc. Following the example went very smooth until running the function sEstUStarThresholdDistribution() where I got the message "Error: could not find function "sEstUStarThreshold-Distribution". This is probably a minor trivial issue in my implementation that can be solved.

**AC**: Thanks for noting. `sEstUStarThresholdDistribution` is a method of the EddyProc class, not a function, and is called as in example B2. In a revised version we

will update the line in B5 to explicitly include the class:

`EddyProc.C$sEstUStarThresholdDistribution(...)`.
* * *

---

## Author Response (AR1)

**Response to reviewers for REddyProc BGD paper**

immediate

**Abstract.** Dear editor,

We submit a revised version of our manuscript "Basic and extensible post-processing of eddy covariance flux data with REddyProc".

First, we provide a point-by-point response to the reviews. We repeat each reviewer's comment (RC) before each of our author's comment (AC) replies in blue.

The most important changes are

– restructuring of package dependencies (see RC1-4) and

– shortening of the manuscript by moving parts of the original manuscript to a new online supplement (see RC1-2).

Next, we provide a marked-up version of the manuscript, where deleted parts show red strike-through and added parts show blue wiggly-underlined. Note that in addition to the marked changes, also figures changed axis labels.

**1  Response**

**1.1  Response RC1**

AC: We thank referee 1 for his constructive comments.

**RC1-1**: The paper adds to a growing segment of extensive method descriptions for reproducible computational research. While the technical focus is laudable, it is also the reason that the manuscript in its current form somewhat misses the scope of BG ("interactions between the biological, chemical, and physical processes"; https://www.biogeosciences.net/index.html). I imagine this might be part of the reason that the other referees declined. Alternative Copernicus journals like AMT ("techniques of data processing"; https://www.atmospheric- measurement-techniques.net/index.html) or GMD ("statistical models", "technical papers"; https://www.geoscientific-model-development.net/) should provide a much better fit.

AC: While the suggested alternative journals maybe fit the content slightly better, they target an audience primarily interested in data processing or model development. Contrary, we want to target an audience of researchers who *use* EC flux data in their studies. We argue that Biogeosciences is the best open access journal to reach this audience. For instance Biogeosciences Journal recently hosted a special issue on eddy covariance data collected in the Australia and New Zealand, which include also more methodological papers "OzFlux: a network for the study of ecosystem carbon and water dynamics across Australia and New Zealand, 2016". Also a search of the keyword "eddy covariance" in the main text on April 4th, 2018 give back 1620 records. We also looked at the number of views in the open review phase and they were more than 350 (April 4th), supporting the argument that Biogeosciences can be the right audience for the article.

**RC1-2**: Here a few points for consideration in such re-submission: - I suggest shortening the Discussion paper manuscript. It should be straightforward to consolidate 22 pages of heavy methodological detail by about 1/3, and focus on novel aspects.

**AC**: We shortened the manuscript mainly by moving parts to an online appendix. We strive to report the details in some way for reproducibility.

Specifically we summarized section 2.2.3 (MDS) in the general gap-filling section and moved the section including former Fig. 3 to the online supplement. Next, we moved details of the day-time flux-partitioning to an online appendix. Further we moved former Appendices A1 and C to the online supplement.

**RC1-3**: - l. 5: "standard tools available in open source environment for processing high frequency (10 or 20 Hz) data into half-hourly quality checked fluxes". At this time open-source environments for eddy-covariance data processing that actually facilitate open development are only emergent. REddyProc provides a substantial and much appreciated contribution to this movement. I suggest to either substantiate the claim of an abundance of open-source high-frequency data processing environments through providing examples, or to provide a more differentiated overview.

**AC**: We agree with the reviewer that open source tools for eddy covariance processing are only now emerging. There are now available some open source tools for processing of high frequency data, for instance the Eddy Pro software, which is open source and widely used. In the revised version we included a footnote with a link to a web-page that lists such packages both in the abstract and the introduction.

**RC1-4**: - l. 8: While it is true that R is a cross-platform language, this does not mean that research is reproducible by using an R-package across platforms. Known as "dependency hell", installing e.g. REddyProc on a standard Debian Linux distribution requires the co-installation of several operating-system-side libraries (libudunits2-0, libudunits2-dev, udunits-bin, libnetcdf-dev) and even more R-side dependencies (backports, praise, evaluate, highr, mlegp, logitnorm, ncdf4, RNetCDF, minpack.lm, segmented, rprojroot, testthat, knitr). In some operating systems such as Windows, there is hardly any automation available for resolving operating-system-side dependencies, making R-packages with heavy dependencies inaccessible to less experienced users. Most importantly, dependency resolution itself is not reproducible among operating systems, thus rendering reproducible research impossible. A balanced discussion of how REddyProc can be used for reproducible research alongside examples for dependency resolution would add much substance and usability to the manuscript.

**AC**: The library dependency issue were caused by requirements of system libraries with the two NetCDF packages. In an updated package version we 1) moved NetCDF related functionality to an own package REddyProcNCDF and 2) migrated function from package logitnorm directly to REddyProc. Hence, in the updated package, besides the suggested packages mainly used in generating documentation, only two dependencies are there, specifically the tidiverse package suite and the mlegp Gaussian process package. The first one is almost as standard as base R, and at least one package dependency will be required if we want ot use Gaussian processes.

Furthermore, we provide automatically-built docker images, a kind of lightweight virtual machine, on docker hub, so that with docker installed, REddyProc can be used without installation of R software and other package dependencies with a single command. In the revised version we only slightly extended the section on installation (Appendix A, p21) but rather refer to the REddyProc homepage on github for those technical issues.

**RC1-5**: - l 15: It could be pointed out that REddyProc has already been adopted for computational research by the flux community, such as in Metzger et al. (2017). These authors also point to a community solution for "dependency hell", an pre-compile REddyProc alongside its dependencies into compute images that contain a turn-key, reproducible and shareable processing environment.

5     **AC**: Thanks for this suggestion. In the revised version we cited the paper when referring to the provided Docker image (Appendix A, p21L11)

**1.2 Response RC2**

**RC2-1**: The paper describes the R package REddyProc containing tools for post-processing of eddy covariance data with a focus on $CO_2$. The REddyProc package contains tools for reading halfhourly data from different formats, estimating u*

10  thresholds, gapfilling, flux-partioning, data visualisation, and estimation of uncertainties. The results obtained with the routines of the package are compared with other state-of-the art tools. This resulted in no significant differences in the results from the different tools on a montly or annual scale. The REddyProc package contain a more sophisticated way of the treatment of seasons and the possiblity to run and take advantage of multi-year data sets. Discussion paperThe paper provides a good description of the rationale of the package and the diffence in calculation method between this package and earlier available

15  tools. The paper also includes an appendix with example of how to run the routines on a full annual dataset. In general, I find that the paper is well written and documented, and that it can provide a very useful reference for scientist using the package. I have some suggestions for improvements in the specific comment below. Providing this package is very good for the flux reasearch community and by using standardized methods the results from different research groups become much more comparable.

    **AC**: We thank reviewer 2 for his encouraging comments.

20  ### 1.2.1 Specific comments

**RC2-2**: Table 1. I find it a bit lazy to write "uStar" instead of the correct symbol "$u_*$".

    **AC**: In the revised version we use now symbol "$u_*$", aside from referinng to source code, where the multiplication character cannot be used.

**RC2-3**: Figure 2. Caption: I suppose it should be "night-time NEE". The "season–temperature subset" is only understandable

25  after careful reading of the main text.

    **AC**: In a revised version we modified the figure 2 caption to "Concept of the $u_*$-filter for a subset of data from DE-Tha. Night-time NEE at low $u_*$ friction velocities, i.e with low turbulence, is biased towards lower NEE values compared to cases with higher $u_*$, although other environmental conditions are similar. The $u_*$ value below which this effect is considered significant, i.e. $u_*$ threshold (dashed line), is estimated by a moving point method on $u_*$ bins (crosses) across half-hourly records (circles)."

**RC2-4**: p.6, l.12-13: It is not completely clear from the text which time-frame is used for the flagging. What is the chosen aggregation period for a "robust u* estimate" (l. 10).

**AC**: $u_*$ value of each half hour is compared to the threshold determined for the corresponding season. In the revised version we slightly extend the paragraph (P6L16)

Both, the aggregation periods for which different thresholds are established, i.e. seasons, and the aggregation across different seasons is specified by the user. The first defaults to three months, starting in months, 3,6,9, and 12, the second defaults to maximum across one year as described at (P11L24ff). The choice of both, however, can be adapted to the knowledge of the site. We added extended discussion at P14L15 and Appendix B7.

**RC2-5**: p.7, l.8.: Probably this is the first time in the paper the "BGC online tool" is mentioned, and should thus be explained and referenced here.

**AC**: Thanks for noting. In the revised version, we extended the description of the BGC online tool in the benchmarking section (3), include a reference in this earlier method description section (P8L10), consistently refer to the BGC online tool as BGC16 (also in variable subscripts in figures), and added this acronym to Table 1.

**RC2-6**: p.9, l.22: The abbreviation "LRC" should be explained.

**AC**: Thanks for noting. In a revised version, we will keep the description of the Light response curve (LRC) in the specific day-time partitioning section (2.3.2 page 9), but write in this earlier general section (P8L24) "fits a model to observations of daytime NEE and global radiation"

**RC2-7**: p.12, l.13: The choice of the software used for comparison (here called "benchmarking") should be explained better. The u* filtering is compared to Papale et al. (2006) and the gap-filling and flux partitioning to the BGC online tool. Do these represent (previous) state-of-the-art or have exactly those routines been used for calculating the FLUXNET data-sets?

**AC**: The required more detailed description of the role of the benchmark tools was described in the more specific sections, 3.2 and 3.5. In the revised version we keep some details in the specific sections but also extended the general description at beginning of the benchmark section (p10L20).

**RC2-8**: p.14, l.6: Should read "To explore these consequences ..."

**AC**: Thanks for noting the typo that is now corrected (P11L16)

**RC2-9**: Discussion paper Fig.4+5: I find it quite interesting that there seems to be a fairly large scatter in the estimated u* thresholds, but this does not seem to translate into a similar scatter in the annual NEE estimates. It would be valuable to have a further discussion on this. Does it e.g. mean that the value of the u* threshold is not very important? A lower u* threshold means that more data are kept which could potentially lower the uncertainty on the annual estimate.

**AC**: We also did not expect such low sensitivity of the aggregated flux results to the differences in the $u_*$ threshold estimates, and discussed this at sections 3.2.3: "indicates that the sensitivity of NEE to the $u_*$ threshold estimate in the inferred ranges is low, which also explains the large uncertainty of the $u_*$ threshold estimate".

In the revised version we elaborated a bit more on this point (section 3.2. on page 14)

**RC2-10**: Fig. 6: The symbols of the quality flags are very difficult to read and distinguish. The legend refers to the sub-figures as "top" and "bottom". Here it should be "left" and "right".

**AC**: In the final (as opposed to review) two column mode formatting, the labels "top" and "bottom" are the correct ones.

The symbol transparency is a compromise between reading a single symbol and conveying the message that most of the values overplot at the 1:1 line. In a revised version we adjust the figure towards less transparency.

**RC2-11**: Appendix B: I tried out the REddyProc package on my MacBook, First I tried to update all packages and after a little tweaking I managed to load REddyProc. Following the example went very smooth until running the function sEstUStarThreshold­Distribution() where I got the message "Error: could not find function "sEstUStarThresholdDistribution". This is probably a minor trivial issue in my implementation that can be solved.

**AC**: Thanks for noting. `sEstUStarThresholdDistribution` is a method of the EddyProc class, not a function, and is called as in example B2. In the revised version we updated the line in B5 to explicitly include the class: `EddyProc.C$sEstUStarThresholdDistribution(...)` and explicitly refer to section B2.

[revised manuscript text omitted]

---

## Author Response (AR2)

Dear Editor,

thank you for recommending the revised manuscript for publication. In this document we address the minor revisions suggested by Referee #3 and provide the manuscript with marked changes below.

*Wutzler and others demonstrate the REddyProc package. I feel that the manuscript will be an important contribution to the flux community given the number of users who will likely use the package.*
Thank you.

*Abstract was a bit qualitative, numbers would help. For example, how much lower and higher are the uncertainty estimates noted toward the end of the abstract?*
We updated the abstract:
Lower uncertainty estimates of both u $*$ and resulting gap-filled fluxes by 50% with the presented tool was achieved by …
Higher estimates of uncertainty in day-time partitioning (about twice as high) resulted

*'allowed a boost in' is too wordy.*
We replaced by „greatly advanced"

*'and temperature' instead of 'or temperature' on line 7 page 2.*
We corrected as suggested.

*Figure 1 already includes a thumbprint plot for RE that seems a bit implausible. Is the red/orange horizontal line through the middle of the plot really true?*
The "reddish" band does not come from "gap filled" NEE estimates but the band coincides from mainly measured data where there is a very high NEE at nighttime (if you look the nighttime NEE immediately before the big gap looks quite high). Hence its in the original measurement data. The color scale is assigned so that the maximum value of the dataset gets the red color code.
We decided to not discuss this distracting detail in the paper.

*The legend of Fig. 2 is a bit elusive but can be figured out with careful study. Clarifying it would help.*
We reworded the figure caption as follows:
Concept of the u$*$-filter: Night-time NEE at low u $*$ is biased towards lower NEE values compared to cases with higher u$*$. Unbiased NEE should scatter around the same plateau because environmental conditions are similar. The u $*$ threshold (dashed line), i.e. the value below which this bias is considered significant, is estimated by a moving point method on u $*$ bins (crosses) across half-hourly records (circles). The example uses a subset data from DE-Tha.

*P 5 and elsewhere: please use 'van Gorsel'. Regarding section 2.1, will improved approaches following van Gorsel et al. (2007), sigma_w-based approaches (Jocher et al., see earlier work by Acevedo: https://www.sciencedirect.com/science/article/pii/S0168192308001962), estimates of drainage (see Hayek et al., 10.1016/j.agrformet.2017.12.186), atmospheric stability approaches (Novick et al., 2004) or other non-ustar approaches be implemented? It's been my feeling that a large part of the flux community has moved beyond ustar.*
We apologize for not correctly citing the two-word surname and corrected it. Currently there are no plans to include more recent approaches. The issue tracker on the ReddyProc github site is a good place to suggest, argue for, and detail such a suggestion.

*2.1.1: I've long-wondered what is the best threshold for 'nighttime' especially given the characteristic decline in turbulence with the collapse of the daytime boundary layer may occur earlier. Is there a justification for Rg < 10 W m-2? (see also the last line of page 8).*

We selected this threshold and other similar values because we had not reason to deviate from the standard approaches, here DP06. We did not discuss this issue in the manuscript but give some thoughts here in the replies.

The night-time threshold for u* estimation 10 Wm-2 as used by Papale et al 2006 was selected to our knowledge mainly to 1) guarantee a certain amount of nighttime data for the estimation of E0 and Rb (this we think is the most important reason); and 2) to avoid problems in the selection of nighttime data in sites with drift in the pyrometer (i.e. positive values at nighttime due to wrong calibrations or drift that were pretty common in the past, now we think that the quality control is stricter in this regard). Also for the point 2 the crosscheck with the potential radiation is important.

Further, the Rg measurement can be a little unclean and therefore small values below 10 are included, for example bright moon hours at night, slight offsets of the instrument, or a not fully dark sky due to some back scatter from low hanging clouds after sunset. For the partitioning, the algorithm additionally even filters for nighttime by calculating the potential radiation and only "true" nighttime values will be considered.

Layering might already set in before, i.e. at higher radiation values like 30 W m-2 and u* estimation and filtering could include even more data points. We are not aware of any study on this and suspect a good threshold to be very site specific and maybe even season specific - for example thinking of hours with low sun angles in Northern sites. However, for this, REddyProc allows the PIs and users to experiment with their own Rg thresholds.

*Regarding 2.2.2, often alternate micrometeorological information is available. Is it possible in REddyProc to add multiple micrometeorological measurements of the same variable?*

We do not fully understand the request. E.g. „Using PAR when Rg is missing" is currently not implemented. However, the user can flexible define different meteorological variables and thresholds with the Lookup-table approach. However, then it is not "the" MDS algorithm any more. There is also currently no feature to use a different temperature column aside from air temperature yet. Again, the github issue tracker would be a good place to propose and details such a feature.

*On page 10, BGC16 was used and defined before. Please define it at first instance.*

Section 3 is dedicated to describe the benchmark tools. When using acronym BGC16 before, its difficult to describe it sufficiently. Hence those usages refer to section 3 (e.g. Table 1 and p8l10).

*'Papale C-implementation' on p. 13, just use DP06. A quick re-read for minor issues like this would improve the paper. Also, what is a 'big scatter' on page 13?*

We hope that a re-read captured those minor issues, as we corrected on page 13.
We reworded „Big scatter" by „although individual threshold estimates differed"

*Regarding the LRC fits, is a least squares cost function applied or least absolute deviation to avoid the influence of outliers (see Richardson 2005/6 papers). Some discussion of the cost function is necessary in my opinion.*

The fitting for least absolute deviation is appropriate if measurement error is Laplace distributed, which seems to be the case both for soil respiration and NEE data. However, when comparing fluxes at similar environmental conditions and hence, of similar magnitude, the distribution is better approximated by a Gaussian distribution, and the apparent Laplace shape is probably generated by a superposition of several Gaussian distributions, where standard deviation scales with flux magnitude (Lasslop 2008). Hence, in accordance with Lasslop 2010 we used the usual normal assumption in fitting the LRC curve.
We extended the section on fitting the LRC curve and repeated the cost equation from Lasslop et al. in supplement 1 (section 2 , step 3).

[revised manuscript text omitted]

**1   Details of marginal distribution sampling (MDS) for gap-filling**

The marginal distribution  (section 2.2.3) combines the two methods look-up-table (LUT) and  mean diurnal course (MDC). Depending on the availability of the meteorological data, three different conditions are identified for each half-hourly NEE flux:

1. The data of the three meteorological variables (Rg, Tair, and VPD) are available.

2. Tair or VPD are missing, but Rg is available.

3. Also Rg is missing.

Case 1): The missing value is replaced by the average value under similar meteorological conditions with respect to Rg, Tair and VPD in a LUT approach. If no similar meteorological conditions (minimum of two half-hourly fluxes) are present within the starting time window of 7 days, the  window size is increased to 14 days.

Case 2): The same LUT approach is taken, but similar meteorological conditions can only be defined via Rg within a time window of 7 days.

Case 3): The missing value is replaced with the mean diurnal course (MDC). The number of days start with one day, thus a linear interpolation of available data at adjacent hours (±1 hour) at the same day. The number is then increased to ±1 and ±2 days.

If after these steps the NEE values could not be filled, the procedure is repeated with increased window sizes until the value can be filled, see flow diagram in Fig. 1.

The provided quality flag depends on both, the number of meteorological conditions present, $n_c$, and the number of days in the window, $n_d$. When using LUT,   the flag equals 1 when $n_c$ is either 1 or 3 and $n_d \leq 7$. It equals 3 if either $n_c = 3$ and  $n_d > 28$, or $n_c = 1$ and $n_d > 14$. In all other LUT cases When using MDC, i.e. $n_c = 0$, the quality flag equals 1 if it is the same day, it equals 3 if $n_d > 2$, and it equals 2 in all other cases.

The three default variables and margins are Rg with ±50 Wm$^{-2}$, Tair with ±2.5 °C, and VPD with ±5.0 hPa. These can be also be specified by the user.

**2  Details of daytime flux-partitioning**

This section reports details of the steps  for daytime flux-partitioning (section 2.3.2).

[Figure]

**Figure 1.** Flow diagram of the MDS gap-filling algorithm as implemented in REddyProc. See table 1 in the main paper for abbreviations.

In step 1, parameter $E_0$ is estimated for 12 day windows. Only records with temperature above -1 °C are valid for estimation. Reference temperature $T_{Ref}$ in eq. 1 of the paper is set to the median temperature of the window in order to decrease correlation between estimates of $R_{Ref}$ and $E_0$. A missing estimate is reported for non-valid windows with too few valid records (minNRecInDayWindow = 10), non-convergence of the fitting procedure, or an $E_0$ estimate outside the bounds [50,400]. Missing estimates are filled during the smoothing in step 2.

In step 2, the Gaussian Process takes into account the uncertainty of the $E_0$ fit from night-time in each window. However, if the correlation of $E_0$ across subsequent windows is high, the uncertainty is reduced similar as with repeated measurements. The respiration $R_{Ref}$ for windows where no fit could be obtained is set to the value from the previous valid window. Again, only records with temperature above -1 °C are used.

In step 3, fitting of other parameters is done for each window centered at the same record as the windows of step 1.

By default the fit uses the same weak prior on parameters as BGC16 (Lasslop et al., 2010). The  optimization minimizes a cost function using the "BFGS" method, which is a quasi-Newton method as published by Shanno (1970). The cost function assumes normally distributed model-data-residuals and normally distributed vague priors (eq. 1) (Lasslop et al., 2010, eq. 5).

$$c = \sum_i \frac{(NEE_{pred,i} - NEE_{obs,i})^2}{\sigma^2_{NEEadj,i}} + \sum_j \frac{(\theta_j - \theta_{prior,j})^2}{\sigma^2_{\theta,j}},$$

(1)

where $NEE_{pred,i}$ is computed by the LRC equation, and parameters $\theta = (k, \alpha, R_{Ref}, \beta)$ have prior locations of $k = 0.05$, $\alpha = 0.1$, $R_{Ref}$ = night-time estimate, and $\beta$ = range of NEE values, i.e. the difference between 97% and 3% quantile. The prior uncertainty is  $\sigma_k = 50$, $\sigma_\beta = 600$, $\sigma_\alpha = 10$, and $\sigma_{R_{Ref}} = 80$.

 The uncertainty of NEE by default is assigned a lower bound (eq. 1). This lower bound differs from Lasslop et al. (2010) and assigns low influence to records with high uncertainty  in order to avoid the problem of  high leverage of a few records with very low estimates of NEE-uncertainty.

$$\sigma^2_{NEEadj,i} = \max[\sigma^2_{NEE,i}, q_{0.3}(\sigma^2_{NEE})], \tag{2}$$

where $q_{0.3}(\sigma^2_{NEE})$ is the 30% quantile of the vector of estimated standard deviations of NEE.

[revised manuscript text omitted]
 $u_*$ thresholds (Fig. 4). The uncertainty estimated by REddyProc was only half as high as the one estimated by DP06. It resulted from acknowledging the seasons of similar conditions also during bootstrap. The lower $u_*$ threshold uncertainty propagated to the uncertainty estimates of NEE (Fig. 5).

**4.2 Gap-filling**

The evaluation of annual aggregated NEE data obtained with REddyProc and BGC16 showed good agreement across sites.

Note, that the aggregated NEE data contained both, measured and gap-filled data for the purpose of evaluating the impact of the processing on the aggregated NEE. The determination

coefficient ($R^2$) showed a very good agreement between the two methods both at annual and monthly time scale (Table 1). The relative mean absolute error (RMAE) is low: about -3 % for annual aggregation and -0.47 % for monthly aggregation. Both Modeling Efficiency (EF) and Mean Bias Error (MBE) showed a very low bias between the two products for both monthly and annual aggregations (Table 1).

**Table 1.** Gap-filling evaluation statistics of yearly and monthly cumulated and gap-filled NEE data obtained with `REddyProc` and BGC16. Statistic abbreviations are explained in the paper's benchmarking section

|        | Yearly | Monthly |
|--------|--------|---------|
| N      | 25     | 281     |
| pearson | 0.99  | 1.00    |
| MBE    | -0.02  | 0.00    |
| RMBE   | 2.8%   | -0.08%  |
| MAE    | 0.02   | 0.00    |
| RMAE   | -3.0%  | -0.5%   |
| RMSE   | 0.09   | 0.01    |
| $R^2$  | 0.98   | 1.00    |
| EF     | 0.98   | 1.00    |

**4.3 Night-time partitioning**

The results show good agreement between $R_{eco}$ estimated using `REddyProc` and BGC16 for $R^2$ and modelling efficiency (Table 2). The relative RMSE is 6.56 and 13.45 % for yearly and monthly aggregation, respectively. The high RRMSE is due to few site years as reported in Table 3, and this is confirmed by the lower RMAE (3.87 % and 6.33 % for yearly and monthly, respectively), which is less sensitive to outliers.

The difference in $R_{eco}$ related to the selection of night-time data are not negligible: the differences in RRMSE of 1.03% and negligible differences in

$$R^2$$

. Also, the use of $E_0$ prescribed from BGC16 lead to negligible difference in $R^2$ of about 0.02. Therefore, though very small, the selection of night-time data is the most important difference introduced by `REddyProc` .

**4.4 Day-time partitioning**

The time-variable estimate of temperature sensitivity of ecosystem respiration with day-time partitioning is a significant source of uncertainty for gross fluxes GPP and $R_{eco}$. `REddyProc` accounts for the previously unaccounted uncertainty for estimating uncertainty of these gross fluxes by a bootstrap (Fig. 6). The BGC16 estimate of annual uncertainty in Fig. 6 is a low estimate compared to a full quantification,

**Table 2.** Nighttime partitioning evaluation statistics across sites of annually and monthly aggregated ecosystem respiration ($R_{eco}$) estimated with `REddyProc` and BGC16.

|        | Yearly | Monthly |
|--------|--------|---------|
| N      | 25     | 297     |
| pearson | 0.99  | 0.99    |
| MBE    | 25.4   | 2.15    |
| RMBE   | 2.2%   | 2.2%    |
| MAE    | 44.8   | 6.17    |
| RMAE   | 3.8%   | 6.3%    |
| RMSE   | 75.6   | 13.2    |
| RRMSE  | 6.5%   | 13.5%   |
| $R^2$  | 0.99   | 0.98    |
| EF     | 0.99   | 0.97    |

|        | RMBE  | RMAE  | RRMSE | $R^2$ | EF    |
|--------|-------|-------|-------|-------|-------|
| CA-NS7 | 15.12 | 16.29 | 32.64 | 0.96  | 0.86  |
| CA-TP3 | -1.99 | 25.10 | 38.59 | 0.87  | 0.82  |
| DE-Hai | 6.00  | 6.48  | 10.40 | 0.99  | 0.96  |
| DE-Tha | 2.83  | 3.52  | 6.19  | 0.99  | 0.99  |
| DK-Sor | 3.50  | 4.45  | 7.34  | 1.00  | 0.99  |
| ES-ES1 | 1.68  | 3.34  | 4.59  | 0.93  | 0.92  |
| ES-VDA | 3.96  | 9.98  | 15.93 | 0.95  | 0.93  |
| FI-Hyy | 2.72  | 3.05  | 4.47  | 1.00  | 1.00  |
| FI-Kaa | -4.52 | 5.51  | 9.07  | 0.99  | 0.99  |
| FR-Gri | 3.96  | 4.14  | 6.35  | 1.00  | 0.99  |
| FR-Hes | 4.11  | 5.06  | 9.06  | 0.99  | 0.99  |
| FR-Lq1 | -2.89 | 5.76  | 7.24  | 0.99  | 0.99  |
| FR-Lq2 | -8.27 | 9.49  | 11.49 | 0.99  | 0.97  |
| FR-Pue | 0.65  | 1.27  | 1.55  | 1.00  | 1.00  |
| IE-Dri | -0.03 | 2.41  | 3.24  | 1.00  | 1.00  |
| IL-Yat | 1.66  | 2.89  | 3.72  | 0.99  | 0.99  |
| IT-Amp | 24.34 | 29.38 | 45.33 | 0.91  | 0.48  |
| IT-MBo | 2.03  | 3.00  | 4.49  | 1.00  | 1.00  |
| IT-SRo | -1.46 | 1.86  | 2.10  | 1.00  | 0.99  |
| PT-Esp | 9.06  | 12.08 | 19.94 | 0.62  | 0.45  |
| RU-Cok | -0.76 | 28.54 | 38.98 | 0.27  | -0.07 |
| SE-Nor | -1.05 | 2.09  | 3.62  | 1.00  | 1.00  |
| US-Ton | 2.93  | 7.45  | 11.82 | 0.93  | 0.92  |
| VU-Coc | 0.73  | 2.64  | 3.83  | 0.93  | 0.93  |

**Table 3.** Nighttime partitioning evaluation statistics at sites level of the of ecosystem respiration ($R_{eco}$) estimated with `REddyProc` and BGC16.

because it assumes no correlation between half-hourly errors. An improved quantification of correlations requires the full variance-covariance matrix of the LRC parameter fits (Lasslop et al., 2010; Menzer et al., 2013), which were not available for BGC16.

The introduced uncertainty is reduced by smoothing the temperature sensitivity estimates ($E_0$) across several successive windows (Fig. 6 7 top) before estimating parameters of the LRC. This smoothing has also an effect on predicted half-hourly gross fluxes (Fig. 7 bottom).

[Figure]

**Figure 6.** Density plot of estimated standard deviation of uncertainty of the annually aggregated GPP across sites due to uncertainty in parameters estimation during day-time based flux partitioning. Higher estimates with `REddyProc` are caused by taking into account the uncertainty in temperature sensitivity, $E_0$.

Results of daytime partitioning are sensitive to subtle details of the procedure. Hence, there is quite much scatter introduced by the differences of `REddyProc` processing with
5 default options or with options that maximize compatibility with BGC16 (Fig. 8 top). One such a subtle options is to decrease or not to account for the unreasonably high leverage of some observations during the fit to the light-response curve by some records having a very small estimate of its uncertainty
10 (Fig. 8 bottom). Due to the sensitivities of the day-time partitioning, there are still differences between `REddyProc` with compatibility options and BGC16 (Fig. 9).

[Figure]

**Figure 7.** Effects of smoothing (top) successive estimates of temperature sensitivity, $E_0$, on predicted GPP (bottom) for site PT-Esp.

[Figure]

[Figure]

**Figure 9.** Histogram of difference between monthly GPP predictions of `REddyProc` with Lasslop10 compatibility options BGC16.

**Figure 8.** Sensitivity of estimated monthly GPP fluxes to specific processing details results in scatter between GPP predictions based on different `REddyProc` options. Most of the differences between default options and compatibility options (top) are caused by differences in weighting different records during the fit (bottom.)